# NeMal: Never-ending Marine Learning - Unleashing the Power of Controllable Image Synthesis for Promoting Marine Visual Understanding

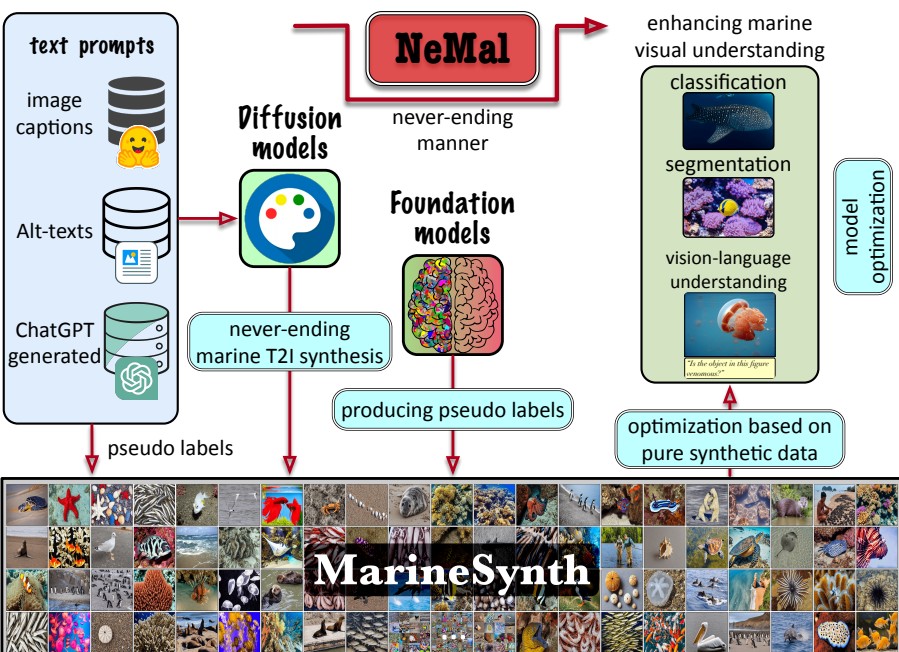

Figure 1: We present **NeMal**, a never-ending marine learning system, to ceaselessly synthesize marine images under text conditions to optimize models for enhancing marine visual understanding based on purely synthetic data. **MarineSynth** produced by NeMal has 4 million meticulously constructed text prompts and corresponding text-to-image (T2I) synthesis outputs, which significantly reduces the human efforts on both data collection and labeling.

## Abstract

The relentless pursuit of marine learning is required by the essential need to understand and protect the complex marine ecosystems that cover over 70% of the surface of our planet. Due to the specific underwater/marine environments, the data collection and labeling are expensive and labor-intensive, also limited to user groups with special equipment. Existing marine visual learning just optimizes models from a small set of marine data with human labels, which cannot fit the essence of ongoing marine exploration. In this work, we propose NeMal, a Never-ending Marine Learning system that harnesses controllable image synthesis and efficient foundation models to perform never-ending marine visual synthesis and understanding. Based on NeMal, we produce MarineSynth, which is the first large-scale marine synthetic dataset to date, featuring more than 4 million unique text prompts and corresponding text-to-image outputs with pseudo labels from text prompts or foundation models. The experiments on downstream classification, segmentation, and vision-language understanding tasks demonstrate the promise of utilizing synthetic data to promote marine visual understanding, significantly reducing human efforts in both data collection and labeling.

# 1 INTRODUCTION

Marine learning is an endless journey. On one hand, marine ecosystem (Epstein et al., 1993; Halpern et al., 2008), being the most productive among all ecosystems, holds significant ecological, social, and economic value. On the other hand, the mystery (Macreadie et al., 2018) of the ocean has inspired continuous researchers to delve into its vast depths, seeking to unveil its secrets and understand its treasures. However, unlike easily available in-air visual data, collecting real-world underwater/marine images for monitoring and understanding marine ecosystems is much more challenging (Williams, 2012) due to its specific underwater environments. The marine visual data is thus relatively small and with intrinsic long-tailed distribution compared with our everyday visual data (Deng et al., 2009; Lin et al., 2014). Existing dominant marine visual studies usually involve describing and analyzing the collected images/videos based on *in-situ* surveying approaches (Biard et al., 2016). There are two main constraints within this line of research: **limited data scalability** (Hollinger et al., 2012) and **annotation costs** (Kohler & Gill, 2006; Beijbom et al., 2015; Pizarro et al., 2017). Due to the significant cost of collecting marine visual data (*e.g.*, several quadrat surveying images (Trygonis & Sini, 2012) require several hours) and the further annotation procedure is usually time-consuming (Beijbom et al., 2015) with expertise involvement, existing marine visual analysis algorithms (Fan et al., 2020; Hong et al., 2023; Varghese et al., 2023; Katija et al., 2022) are limited to few pre-defined marine conceptions (Lian et al., 2023) or scenarios (Beijbom et al., 2015). We believe that these algorithms lack both the richness and scalability required for gathering massive amounts of marine visual knowledge and performing **continuous exploration and study** (Chen et al., 2013; Mitchell et al., 2018) of marine ecosystems.

We have recently been witnessing great success led by foundation models (Li et al., 2022; 2023a; Kirillov et al., 2023; Liu et al., 2023b; Zhu et al., 2023), driven by a significant scale of training data (Shao et al., 2019; Zhou et al., 2017; Gupta et al., 2019) and powerful networks (Zhang et al., 2022; Carion et al., 2020; Dosovitskiy et al., 2020). Such foundation model recipe leads to efficient and flexible models, supporting a wide spectrum of downstream tasks. Concurrently, text-to-image (T2I) synthesis (Wang et al., 2024a; Hu et al., 2024; Zhou et al., 2024) has also gained remarkable attention due to its impressive controllable image generation performance. Among various generative models, diffusion models (Rombach et al., 2022a) are popular for their high-quality generation capabilities following text conditions. The readily paired text prompt and corresponding synthesis output have stimulated several works (Nguyen et al., 2024; Feng et al., 2024; Li et al., 2023b) to build synthetic datasets for model optimization with minimal human efforts.

There are a few attempts (Xie et al., 2009; Dhurandher et al., 2008; Potokar et al., 2022) at generating synthetic data that have been explored in the marine field to address the scarcity of labeled data. However, the synthetic images from the underwater simulator (Potokar et al., 2022) struggle with the diversity and coverage. Meanwhile, it also requires non-negligible human efforts to build marine scenarios in advance, indicating that collecting diverse synthetic images from simulators cannot be scalable. In this work, we consider adopting powerful T2I models (Rombach et al., 2022a; von Platen et al., 2022; Ruiz et al., 2023) as the surrogate to perform continuous image generation under the text conditions as illustrated in Fig. 1, which exploits gathered knowledge in the trained models to generate diverse marine images consistent with real-world images. This brings a significant novel advantage, unprecedented in existing algorithms: we can generate data with conditions at any scale as a never-ending learner, arbitrarily increasing the volume of synthetic data with little human intervention. To ensure the diversity and faithfulness of text prompts, we first construct our marine conception list to get a balanced data distribution and then utilize powerful large language models (OpenAI, 2022; Jiang et al., 2024) to generate/rewrite the text prompts to make them adhere to real-world constraints. After T2I synthesis, we pair the synthesized images with pseudo labels (from text prompts or foundation models for task-specific supervision) for model optimization to enhance marine visual understanding. Theoretically, we could achieve **never-ending marine learning** by streamlining LLM for continuous text prompt generation, T2I synthesis for scalable data generation, pseudo label generation from text prompts or foundation models, and model optimization based on paired image supervision.

We demonstrate the advantage of our NeMal in Fig. 2. We take classification as an example and pseudo labels of synthetic data are from text prompts. Considering the above-mentioned challenges on real data collection and labeling, there are only limited real-world data available in our case. As illustrated, there are some tiny gaps between models optimized by real data and synthetic data. By continuously generating required images following the conditions, we could achieve competitive

or even better classification performance based on purely synthetic data by scaling up the synthetic images. Obviously, combining synthetic data and real data together archives the best recognition performance with promising performance gains over the settings of solely utilizing real data or synthetic data. Finally, NeMal is not without an upper bound as illustrated in Fig. 2, which is subject to both 1) the ability of the generative models to generate aligned and faithful images as the given text prompts; 2) the quality of pseudo labels from the text prompts (they may have hallucinations) or foundation models (wrong predictions). We also observe that we can improve the upper bound by increasing the size of real data and, meanwhile, better alleviate the influence of limited labeled real data (*e.g.*, 20-shot vs. 40-shot) by continuously increasing synthetic data.

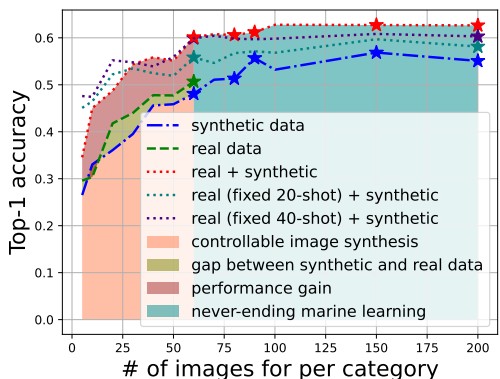

There are two similar works with our NeMal: NEIL (Chen et al., 2013) and NELL (Mitchell et al., 2018) formulating a never-ending learner for image learning and language understanding, respectively. We clarify two main differences between NeMal and these two works: 1) both NEIL and NELL must access real data to discover common-sense relationships and knowledge in a semi-supervised manner while NeMal learns from synthetic data; 2) NEIL and NELL emphasize the never-ending manner from the perspective that their algorithms could learn the Internet data 24 hours/day while NeMal underscores the continuous generation of synthetic data for model optimization by assembling powerful LLM, T2I model and foundation models.

Figure 2: Illustration of NeMal and we take classification as an example. Best viewed in color.

Furthermore, we do not focus on promoting the image quality of synthetic images or performing data augmentation based on real data as DreamDA (Fu et al., 2024). Instead, we demonstrate the efficiency of utilizing synthetic images to support task-agnostic marine visual understanding (including classification, dense segmentation, and vision-language understanding tasks). Our NeMal serves as a pioneering and invaluable start for utilizing synthetic data for domain research with minimal human efforts on both data collection and data labeling. Our main contributions are summarized as follows:

- We propose NeMal, the first never-ending marine learning system to continuously synthesize faithful marine images following text conditions and perform marine visual understanding based on synthetic data, supporting downstream task-agnostic visual perception.

- We produce MarineSynth with more than 4 million text-image pairs with ignorable human efforts on data collection and labeling, the largest marine synthetic dataset to date.

- We demonstrate promise of utilizing synthetic data to enhance marine visual understanding. NeMal presents a practical and systematic framework to continuously learn from synthetic data.

## 2 RELATED WORKS

### 2.1 FOUNDATION MODELS

Foundation models (*e.g.*, CLIP (Radford et al., 2021), ALIGN (Jia et al., 2021) and SAM (Kirillov et al., 2023)) have been widely favored by the whole research community. SAM, optimized by billions of masks, demonstrates a strong zero-shot mask generation ability on unseen images. Vision-language models (VLMs) (Zhu et al., 2023; Liu et al., 2023b; Zheng et al., 2023; Li et al., 2022; 2023a) bridges vision modality and text modality together to harness the power of large language models (LLMs) (OpenAI, 2022; 2023) and vision encoders (Dosovitskiy et al., 2020). Optimized by millions of image-text pairs, CLIP (Radford et al., 2021) demonstrated a strong zero-shot recognition ability for diverse images. Considering the formidable capabilities of existing foundation models in perception and reasoning, some research works (Liu et al., 2023d; Ren et al., 2024) proposed to assemble powerful foundation models together to decouple complicated vision tasks for accomplishing step-by-step visual reasoning in a training-free model assembly manner, which makes the whole pipeline more efficient and flexible.

## 2.2 T2I SYNTHESIS

Latent diffusion models (LDMs) Ramesh et al. (2022); Rombach et al. (2022b); Podell et al. (2023) formulate the process of T2I generation through iterative denoising steps initiated from Gaussian noise. LDMs (von Platen et al., 2022; Ruiz et al., 2023) have become widely favored, as their compact latent space improves model efficiency. Stable Diffusion (Rombach et al., 2022b) is pre-trained on the massive LAION-5B (Schuhmann et al., 2022) dataset, around 5 billion text-image pairs for the T2I generation task, which leads to a strong generalization capacity on generating diverse images and formulates redundant semantic priors. Layout-to-Image methods (Chen et al., 2024; Xie et al., 2023; Zhou et al., 2024; Wang et al., 2024a;b) extend the pre-trained T2I model to integrate layout information into the generation and achieve instance position controlling. InstanceDiffusion (Wang et al., 2024a) utilized existing foundation models (Zhang et al., 2023; Ren et al., 2024; Liu et al., 2023c) to produce signal-output pairs. However, it struggles to isolate the attributes of multiple instances and suffers from noisy or fully wrong BBOX inputs, thus generating images with error accumulations. Also, how to generate reasonable layouts that adhere to real data distribution is a key prerequisite for high-quality layout-to-image synthesis. Following the groundbreaking works of image synthesis, numerous studies (Nguyen et al., 2024; Feng et al., 2024; Li et al., 2023b; Hammoud et al., 2024) have focused on utilizing diffusion models for synthetic dataset construction. DatasetDiffusion (Nguyen et al., 2024) proposed to utilize the cross-attention feature maps to generate pseudo labels for semantic segmentation. However, these attempts are still limited to general-purpose domains. Our NeMal is the first attempt to utilize controllable image synthesis for marine visual analysis, which requires specific domain knowledge and design.

**Comparisons with existing algorithms**. To better clarify the relationships and differences between NeMal and existing algorithms, we provide a direct comparison between NeMal and most relative algorithms in Table 1. We focus on five different aspects: the use of *pure synthetic data*, *task-agnostic* support, *never-ending learning* ability, *domain-specific* support, and *human efforts* required.

Table 1: Direct comparison between our NeMal and existing algorithms.

| Methods | Pure synthetic data | Task agnostic | Never-ending | Domain specific | Human efforts |
|---|---|---|---|---|---|
| NEIL (Chen et al., 2013) | ✗ | ✗ | ✓ | ✗ | Labelset definition |
| DatasetDiffusion (Nguyen et al., 2024) | ✓ | ✗ | ✗ | ✗ | Pre-defined labelsets |
| DetDiffusion (Wang et al., 2024b) | ✗ | ✗ | ✗ | ✗ | Pre-defined labelsets |
| InstaGen (Feng et al., 2024) | ✗ | ✗ | ✗ | ✗ | Vocabulary list |
| ImageNet-D (Zhang et al., 2024) | ✓ | ✗ | ✗ | ✗ | Fixed categories |
| TrackDiffusion (Li et al., 2023b) | ✗ | ✗ | ✗ | ✗ | BBOX sequences |
| NeMal | ✓ | ✓ | ✓ | ✓ | Conception list |

## 3 APPROACH

**Preliminaries**. We develop a marine T2I synthesizer by fine-tuning the existing Stable Diffusion ("SD1.5") model, generating images by iterative denoising of a random Gaussian distribution. The training of the SD1.5 consists of a forward Markov process, where real data $x_0$ is gradually transformed to random noise $x_T \sim \mathcal{N}(0, I)$ by sequentially adding Gaussian perturbations in $T$ time steps, *i.e.* $x_t = \sqrt{\alpha_t}x_0 + \sqrt{1 - \alpha_t}\epsilon$. The model is trained to learn the backward process parameterized by $\theta$:

$$p_\theta(x_0|\texttt{Text}) = \int \Big[ p_\theta(x_T) \prod p_\theta^t(x_{t-1}|x_t, \texttt{Text}) \Big] dx_{1:T}, \quad (1)$$

where the training objective optimizes the variational lower bound via a simple reconstruction loss:

$$\min_\theta \mathbb{E}_{x_t, t, \texttt{Text}, \epsilon \sim \mathcal{N}(\mathbf{0}, \mathbf{I})} \left[ w_t \|\epsilon - \epsilon_\theta(x_t, t, \texttt{Text})\|_2^2 \right], \quad (2)$$

where the text prompt $\texttt{Text}$ is to condition the generation process. The model is trained to predict the noise added to create the input noisy image $x_t$. During inference, we gradually denoise a random Gaussian noise for a fixed time step $T = 50$. To help generate diverse and faithful marine images with multiple instances within marine images, we have acquired a comprehensive and wide spectrum of real-world underwater/marine text-image pairs (discussed in Appendix) to fine-tune SD1.5.

### 3.1 SYNTHETIC DATA CONSTRUCTION

We formulate the following procedures for continuously generating marine images as shown in Fig. 3: 1) marine conception list construction in Sec. 3.1.1 to ensure coverage, comprehensiveness, and variability; 2) text prompt generation in Sec. 3.1.2 to generate diverse and faithful images since meaningful text prompts could better guarantee that the generated images match corresponding descriptive prompts; and 3) marine T2I synthesis based on our fine-tuned SD1.5 with 4) preference-based image picking in Sec. 3.1.3 that combines human feedback to filter out those unsatisfactory examples and generate more faithful and realistic images.

Figure 3: Dataset construction of proposed NeMal, including 1) marine conception list construction; 2) text prompt generation, and 3) marine T2I synthesis with 4) preference-based image picking.

### 3.1.1 CONCEPTION LIST CONSTRUCTION

The comprehensive marine conception list is necessary to guarantee coverage and variability of the synthesized data. We believe the success of CLIP is mainly from the meticulously constructed conception list (Xu et al., 2023). A comprehensive marine conception list contains a wide range of objects and enforces a balanced representation. Following this recipe, we carefully design our marine conception list from 5 main aspects: *biology*, *engineering*, *science*, *ecosystem*, and *sustainability*. Finally, we have obtained a detailed and comprehensive list with 2,332 different marine conceptions as illustrated in Fig. 4(a). Our meticulously constructed marine conception list could adequately bridge the distribution gap between real-world and synthesized marine images, ensuring diversity, coverage, and faithfulness of generated images. We leave more details and statistics in our Appendix.

### 3.1.2 TEXT PROMPT GENERATION

The text prompts are important for generating diverse and faithful image outputs. Meaningful text prompts ensure that the generated images match corresponding descriptive captions. We construct three sources shown in Fig. 3: 1) image captions of real-world images. We utilize existing VLMs to generate captions for visual images and the generated captions are then utilized as text prompts. 2) Alt-texts are scraped from the Internet and we rewrite the Alt-texts (statistics visualized in Fig. 4(b)) based on LLM (Mistral 8×7B (Jiang et al., 2024)) to generate more appropriate and informative text prompts, following real-world data distribution and constraints. 3) ChatGPT-generated text prompts (OpenAI, 2022; 2023) (query ChatGPT with conception and instructions), which are diverse and scalable but may have some intrinsic hallucinations. The generated text prompts of NeMal are automatically scalable and from different sources, allowing them to achieve better trade-offs.

### 3.1.3 MARINE T2I SYNTHESIS

Our NeMal proposes marine T2I synthesis as a surrogate for never-ending marine learning. We first construct our internal marine text-image data based on our marine conception list for fine-tuning. We then perform marine T2I synthesis to generate synthetic data with constructed text prompts. To further promote the marine T2I synthesis performance, we formulate the *preference-based image picking* to generate more faithful and realistic images. Our preference-based image picking combines human feedback to filter out the unsatisfactory examples:

$$\mathcal{L}_{bin.} = -(y \log(p) + (1 - y) \log(1 - p)), \ p = \texttt{Selector}(x_0), \tag{3}$$

where $y$ indicates the human preference among two synthesized images from the same text prompt. The `Selector` is a frozen binary classifier optimized by **100K** image pairs with human preferences (12 marine biologist volunteers were asked to select a better one from two images synthesized by the same text prompts). The text prompts are the rewritten alt-texts scraped from the Internet to ensure diversity. We perform iterative image synthesis by $m$ times as demonstrated in Fig. 3 to generate more faithful outputs shown in Fig. 4(c), leading to better downstream visual perception performances.

### 3.2 NEVER-ENDING MARINE LEARNING

**Never-ending manner**. We achieve never-ending marine learning by continuously generating synthetic data and optimizing models based on continuously increasing synthetic data with pseudo

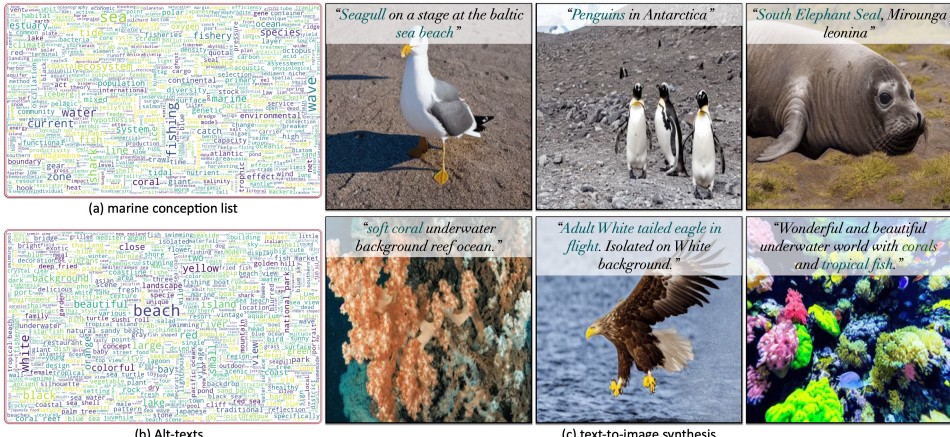

Figure 4: The world visualization of a) constructed marine conception list and b) the top 1000 words of extracted phrases from our rewritten alt-texts. We present marine T2I synthesis results in c).

labels. The pseudo labels for synthetic images are from given text prompts (*e.g.*, category and captions) or existing foundation models (*e.g.*, mask). Please note that the never-ending learning depends on the user requirements and the models could be optimized and fine-tuned constantly since we could continuously generate text prompts (from Alt-texts and ChatGPT) and then synthesize images with pseudo labels (from text prompts or foundation models). In detail, we adopt three representative tasks: *classification*, *segmentation*, and *vision-language understanding* with entirely synthetic image-supervision pairs to substantiate the effectiveness of NeMaL on promoting marine visual understanding. Departing from previous methods relying on real data, we explore the promise and also the upper bound of the synthetic data on enhancing various downstream tasks.

**Promoting marine visual understanding**. We formulate three orthogonal properties for text prompts (as the start point of our NeMaL) on downstream marine vision tasks: 1) **distribution alignment**: the alignment between synthetic data and testing data from downstream vision tasks; 2) **faithfulness**: adhering to real-world data distribution and physical constraints; 3) **generalization ability**: diversity and coverage. We summarize the detailed strengths and weaknesses: the image captions from existing VLMs, must be derived from real images, revealing real data distribution and thus leading to information leakage. Meanwhile, VLMs may also fail to recognize the visual contents accurately, resulting in further data contamination. Alt-texts are diverse but have low distribution alignment with usually downsampled small testing sets since they cover a wide spectrum of information of a specified marine conception. Though ChatGPT can generate redundant text prompts based on various instructions, inaccuracies and deviations from real data are inevitable due to the LLM hallucinations. We leave a more comprehensive and detailed analysis in Sec. 4.2.

## 4 EXPERIMENTS

We perform tailored classification, segmentation, and vision-language understanding experiments. All models are trained by **synthetic** images (from fine-tuned SD1.5 model) and tested by **real** images.

### 4.1 MARINESYNTH DATASET CONSTRUCTION

We construct **MarineSynth**, the largest marine synthetic dataset to date, which contains more than 4 million image-text pairs synthesized in the manner described in Fig. 3. We approach *diversity*, *faithfulness*, and *coverage* by carefully constructing our redundant marine-specific text prompts. The text prompts consist of 1) crowd-sourcing Alt-texts from the Internet by querying meticulously collected marine conceptions and rewriting the scraped Alt-texts based on open-sourced Mistral $8 \times 7B$ to generate more consistent and informative prompts; 2) ChatGPT-generated text prompts based on our marine conception list. We have collected 2 million Alt-texts from the Internet (including WiKi, marine books, and official websites) and 2.3 million ($2,332 \times 1,000$) ChatGPT-generated text prompts. Based on these redundant and diverse text prompts, we utilize our fine-tuned marine-specific SD1.5 model to synthesize corresponding images. We provide more implementation details and dataset statistics on constructing our MarineSynth in our Appendix. All the text prompts and synthesized images will be released to foster utilizing synthetic data for marine research.

## 4.2 MARINE IMAGE CLASSIFICATION

**Testing sets construction**. We start with classification. We adopt the real images from the Sea-animal dataset (ani, 2018) (11,700 images from 23 categories) for experiments. We constructed three testing sets to evaluate the trained models under different settings. The sets contain 1) in-distribution ("**IND**") data, where 100 random images for each category are chosen from the Sea-animal dataset for testing and the rest of the images (9,400 images) are used for optimizing the models as the *Oracle* setting; 2) out-of-distribution ("**OOD**") data scraped from the Internet based on category names, where 100 images for each category are selected and reviewed by humans; 3) human constructed challenging set ("**CLG**" with 2,695 images in total), containing *watercolor*, *cartoon*, *abstract painting*, *artist* and *sketch* images from the same 23 categories. The constructed three testing sets could help better measure the strengthens and limitations of different optimized models.

**Text prompt generation**. We explore the effectiveness of text prompts on downstream classification tasks. For text prompt generation, we consider 3 different sources: 1) image captions generated by VLMs (*e.g.*, BLIP2 (Li et al., 2023a)) by *accessing real-world images* (denoted as BLIP2†); 2) scraped **Alt-texts** from the Internet and we utilize powerful Mistral $8\times7$B to rewrite the alt-texts for generating better and consistent text prompts; and 3) **ChatGPT**-generated text prompts based on the given keywords (*e.g.*, the object category names) while following the captioning style. We generate **500 text prompts** for each object category under different settings and all the classification models have been optimized by using the same hyper-parameters to make a fair comparison. We report all the experimental results in Table 2. The *Oracle* model could achieve high accuracy for the "IND" set, however, struggles with "OOD" and "CLG" sets.

**Pure synthetic data**. Then we perform experiments based on pure synthetic images produced from various text prompts to dissect the gap between models optimized by synthetic data and real data. We have such observations: 1) the image captions generated by BLIP2 could lead to the best classification performance for the "IND" set among all the settings since BLIP2 can access the real-world images from the same data distribution as the "IND" set (potential information leakage), but struggles with poor generalization ability to "OOD" and "CLG" sets. Meanwhile, we cannot ensure the generated captions by BLIP2 can accurately reveal the image contents, thus leading to error accumulation. The limited diversity of generated images presents a formidable challenge in optimizing models, that generalize to unseen data well. 2) Alt-texts possess high diversity, which indicates low distribution alignment with the "IND" set but in contrast strong generalization ability to both "OOD" and "CLG" sets. We also admit that the alt-texts are somewhat noisy or partial alt-texts mismatch the required categories. 3) ChatGPT-generated text prompts share strong generalization ability to the "OOD" set. However, there are still some hallucinations in the ChatGPT-generated text prompts. Besides, some ChatGPT-generated prompts are over-detailed and the SD model fails to generate satisfactory and reasonable images (discussed in the Appendix). 4) Combining ChatGPT-generated prompts and Alt-texts leads to the overall best performance since it combines the strengths of both ChatGPT-generated prompts and Alt-texts. Finally, the models optimized by pure synthetic data produced by our fine-tuned SD1.5 cannot beat the *Oracle* model: 53.66 vs. 57.83.

**Combining real and synthetic data**. Furthermore, we combine the synthetic data with real data under two settings: *Oracle* and *Few-shot* (*e.g.*, "5-shot" indicating each category only has 5 images; and "imbalanced" indicating 4 dominant categories have redundant images while other categories only have 5 images). The experimental results of utilizing few-shot real images are also reported for better comparison. Combining the synthetic data and real data could lead to observable performance gains: around 10 points of improvement (few-shot learning algorithms were compared in our Appendix). We also observe that we can achieve comparable performance with *Oracle* setting under the "5-shot+ChatGPT+Alt-texts" setting (57.83 vs. 57.25), which demonstrates that we could significantly reduce the efforts on both data collection and labeling. The synthetic data could help boost the classification accuracy under all the settings.

**Upper bound of synthesized images**. We first explore the effectiveness of fine-tuning the general-purpose SD1.5 model to the marine domain in Table 3 following the same experimental setting and we included more diffusion models for comparison in Appendix. Please note all the models have been optimized by pure synthetic data. The experimental results demonstrate that our fine-tuning could help generate better marine images that adhere to real-world data distribution, leading to better classification performance. Moreover, we provide the top-1 accuracy curve of utilizing different numbers of synthetic images (synthesized by ChatGPT-generated text prompts) for each category

Table 2: Classification results (Top-1 accuracy, higher is better) of different models optimized under various settings (500 synthetic images used for each category).

| Settings | ResNet-18 | | | | ResNet-50 | | | |
|---|---|---|---|---|---|---|---|---|
| | IND | OOD | CLG | Avg. | IND | OOD | CLG | Avg. |
| Oracle (pure real data) | 74.65 | 56.91 | 36.29 | 55.95 | 75.82 | 59.30 | 38.37 | 57.83 |
| *Pure synthetic data* | | | | | | | | |
| ChatGPT+Alt-texts | 49.26 | 59.87 | 45.12 | 51.42 | 53.11 | 61.48 | 46.38 | 53.66 |
| ChatGPT | 46.48 | 54.96 | 35.84 | 45.76 | 52.98 | 59.48 | 41.97 | 51.48 |
| Alt-texts | 43.87 | 57.43 | 42.37 | 47.89 | 48.06 | 57.30 | 47.87 | 51.08 |
| BLIP2† | 53.43 | 51.22 | 34.84 | 46.50 | 54.37 | 49.91 | 32.65 | 45.64 |
| *Pure synthetic data + Oracle* | | | | | | | | |
| ChatGPT + Alt-texts + Oracle | 75.43 | **70.83** | **49.17** | **65.14** | 77.30 | 70.52 | **53.58** | **67.13** |
| ChatGPT + Oracle | 74.61 | 70.35 | 47.16 | 64.04 | **77.65** | **72.30** | 50.80 | 66.92 |
| Alt-texts + Oracle | 74.87 | 68.48 | 45.64 | 63.00 | 76.78 | 69.61 | 46.27 | 64.22 |
| BLIP2† + Oracle | **75.83** | 62.22 | 40.15 | 59.40 | 76.87 | 66.17 | 43.19 | 62.08 |
| *Few-shot real data* | | | | | | | | |
| 5-shot | 34.52 | 23.48 | 14.73 | 24.24 | 35.62 | 25.13 | 16.10 | 25.62 |
| 10-shot | 42.74 | 30.91 | 19.37 | 31.01 | 44.89 | 30.14 | 18.85 | 31.29 |
| 20-shot | 49.70 | 38.22 | 22.23 | 36.72 | 53.20 | 39.04 | 23.41 | 38.55 |
| Imbalanced | 35.57 | 27.13 | 16.99 | 26.56 | 37.80 | 28.74 | 19.11 | 28.55 |
| *Few-shot real data + synthetic data* | | | | | | | | |
| 5-shot + ChatGPT + Alt-texts | 56.66 | 58.70 | 44.94 | 53.43 | 58.98 | 65.17 | 47.61 | 57.25 |
| 5-shot + ChatGPT | 52.43 | 55.74 | 38.48 | 48.88 | 56.33 | 57.74 | 39.00 | 51.02 |
| 5-shot + Alt-texts | 57.04 | 61.91 | 44.12 | 54.36 | 59.24 | 61.61 | 47.83 | 56.23 |
| 5-shot + BLIP2† | 56.13 | 50.87 | 33.36 | 46.79 | 58.03 | 51.09 | 34.77 | 47.96 |
| 10-shot + ChatGPT + Alt-texts | 60.03 | 62.04 | 43.45 | 55.17 | 62.77 | 62.61 | 47.76 | 57.71 |
| 10-shot + ChatGPT | 56.91 | 57.61 | 37.44 | 50.65 | 60.37 | 58.22 | 39.33 | 52.64 |
| 10-shot + Alt-texts | 56.74 | 57.39 | 42.89 | 52.34 | 61.11 | 63.65 | 47.83 | 57.53 |
| 10-shot + BLIP2† | 59.39 | 51.00 | 34.77 | 48.39 | 62.46 | 54.09 | 34.32 | 50.29 |
| 20-shot + ChatGPT + Alt-texts | 62.16 | 63.30 | 42.97 | 56.14 | 66.46 | 69.26 | 47.72 | 61.15 |
| 20-shot + ChatGPT | 59.63 | 59.57 | 41.22 | 53.47 | 63.16 | 62.13 | 42.30 | 55.86 |
| 20-shot + Alt-texts | 59.65 | 62.65 | 44.42 | 55.57 | 61.98 | 63.83 | 47.20 | 57.67 |
| 20-shot + BLIP2† | 63.39 | 53.09 | 33.43 | 49.97 | 64.64 | 57.91 | 37.22 | 53.26 |
| Imbalanced + ChatGPT + Alt-texts | 56.30 | 62.70 | 45.68 | 54.89 | 60.46 | 61.83 | 42.15 | 54.81 |
| Imbalanced + ChatGPT | 53.74 | 55.78 | 39.55 | 49.69 | 55.72 | 57.74 | 38.81 | 50.76 |
| Imbalanced + Alt-texts | 50.48 | 53.22 | 38.44 | 47.38 | 54.28 | 58.09 | 43.82 | 52.06 |
| Imbalanced + BLIP2† | 53.48 | 49.96 | 34.03 | 45.82 | 55.98 | 50.65 | 36.07 | 47.57 |

Table 3: Ablation studies of effectiveness on fine-tuning SD1.5 to marine domain. Numbers in cyan/magenta/olive/teal indicate the accuracy in IND/OOD/CLG/Avg. settings, respectively. Higher is better and best viewed in color.

| Method | Backbone | ChatGPT | Alt-texts | BLIP2† |
|---|---|---|---|---|
| Vanilla SD1.5 | ResNet-18 | 40.04/51.09 /39.52/43.55 | 38.04/47.96 /39.26/41.75 | 43.04/47.30 /36.44/42.26 |
| Fine-tuned SD1.5 | ResNet-18 | 46.48/54.96 /35.84/45.76 | 43.87/57.43 /42.37/47.88 | 53.43/51.22 /34.84/46.50 |
| Vanilla SD1.5 | ResNet-50 | 42.74/54.87 /39.74/45.78 | 42.35/57.61 /47.16/49.04 | 44.78/49.70 /36.18/43.55 |
| Fine-tuned SD1.5 | ResNet-50 | 52.98/59.48 /41.97/51.48 | 48.06/57.30 /47.87/51.08 | 54.37/49.91 /32.65/45.64 |

Table 4: The coral segmentation results from various segmentation models under different settings.

| Method | Back. | IoU ↑ | Accuracy ↑ | MAE ↓ |
|---|---|---|---|---|
| DeeplabV3 | R50-D8 | 18.73 | 41.57 | 0.3284 |
| SegFormer | Mit-B5 | 42.74 | 65 .24 | 0.2037 |
| SAM♡ | Vit-B | 28.77 | 38.53 | 0.4449 |
| SAM-F♡ | Vit-B | 33.76 | 44.23 | 0.4056 |
| SAM♠ | Vit-B | 44.12 | 52.93 | 0.3688 |
| SAM-F♠ | Vit-B | 46.92 | 58.46 | 0.3194 |
| SAM♡ | Vit-L | 31.78 | 35.97 | 0.5318 |
| SAM-F♡ | Vit-L | 35.50 | 44.18 | 0.3915 |
| SAM♠ | Vit-L | 38.39 | 48.06 | 0.4772 |
| SAM-F♠ | Vit-L | 49.03 | 53.98 | 0.3435 |

Figure 5: **Left**) the scaling observation of using different numbers of synthesized images generated from ChatGPT-generated prompts. **Right**) trials of performing binary image picking, where an appropriate number of trials leads to better classification performance.

on the left side of Fig. 5. With more synthetic images, the top-1 classification accuracy converges, proving that we cannot unlimitedly promote the classification accuracy based on pure synthetic data. We attribute this phenomenon to two reasons: 1) the upper bound of the SD1.5 model on generating photorealistic and required images, accurately conveying the semantics of the given prompt; 2) the massively constructed text prompts are still with hallucinations or noise, leading to noise or artifacts within synthesized outputs.

**Effectiveness of binary image picking**. We provide the accuracy curve of constructing different times of trials for generating corresponding synthesized images (500 synthetic images used for each category) on the right side of Fig. 5. An appropriate number (*e.g.*, 4) could lead to performance gains. Similarly, we cannot achieve unlimited accuracy improvement by solely increasing $m$ even if our binary image picking supports numerous trials. Furthermore, the trade-off between computational costs and performance gains should be considered.

### 4.3 CORAL REEF SEGMENTATION

We then evaluate the effectiveness of synthesized images on dense pixel-level segmentation task, which usually requires high-quality and faithful image synthesis. We choose coral reef segmentation as the proxy task. In detail, we generate 20K coral reef images based on 20K rewritten alt-texts from the Internet and utilize the foundation model CoralSCOP (Zheng et al., 2024) to generate pseudo labels for the synthesized coral reef images. The pseudo labels are paired to optimize various

dense segmentation algorithms (DeepLabV3 (Chen et al., 2017), SegFormer (Xie et al., 2021) and SAM (Kirillov et al., 2023) (denoted as SAM-F)) for coral reef segmentation, discriminating coral reefs from the background. Please note that there is no involvement of real coral reef images or coral mask labeling during the training procedure. At the testing stage, we adopt 400 unseen real-world coral reef images with labeled coral masks by coral biologists for evaluation. We compute the IoU, pixel accuracy, and the mean absolute error (MAE) between model-generated masks and ground truths in Table 4. The results of vanilla SAM are also included for better comparison. For both SAM and SAM-F, we compute the statistics under two settings: "Automatic$^\heartsuit$" (no human prompt is given) and "1 point prompt$^\spadesuit$" (point prompt is given: one random point inside each labeled coral mask). As demonstrated, even without real coral reef images, we could still boost coral reef segmentation performance by assembling controllable image synthesis and foundation models, demonstrating the potential of synthetic data with pseudo labels for domain-specific analysis.

### 4.4 MARINE VISION-LANGUAGE UNDERSTANDING

Finally, we aim to demonstrate our constructed MarineSynth dataset could promote marine vision-language understanding performance on real images. Considering all the synthesized images are paired with the given text prompts, we utilize these image-text pairs to continuously fine-tune the VLMs (MiniGPT4 (Zhu et al., 2023), LLaVA (Liu et al., 2023b) and LLaVa-1.5 (Liu et al., 2023a)). To quantitatively mea-

Table 5: Accuracy (%) of VLMs under different settings. ★ indicates that models have been fine-tuned on our MarineSynth dataset.

| Model | Biology | Engineering | Science | Ecosystem | Sustainability | Avg. |
|---|---|---|---|---|---|---|
| MiniGPT-4 | 67 | 78 | 72 | 79 | 81 | 75.4 |
| MiniGPT-4★ | 71 | 82 | 75 | 83 | 84 | 79.0$_{+3.6}$ |
| LLaVa | 69 | 84 | 75 | 78 | 84 | 78.0 |
| LLaVa★ | 73 | 87 | 79 | 82 | 87 | 81.6$_{+3.6}$ |
| LLaVa-1.5 | 71 | 87 | 76 | 79 | 86 | 79.8 |
| LLaVa-1.5★ | 75 | 89 | 81 | 83 | 91 | 83.8$_{+4.0}$ |

sure the ability of these VLMs to perform marine image comprehending, we construct 500 real-world marine visual question-answering pairs formulated from different aspects: biological species identification, marine engineering fact evaluation, marine science knowledge, marine ecosystem, and sustainability common sense. Each contains 100 pairs. Particularly, following the design of ImageNet-D (Zhang et al., 2024), we construct binary classification to do image-based question answering. We compare the accuracy of original VLMs and the fine-tuned counterparts in Table 5. The performance improvements on both MiniGPT4 and LLaVa demonstrate that we could harness the power of T2I synthesis and VLMs for better marine learning with minimal efforts on data collection and labeling.

## 5 DISCUSSIONS AND CONCLUSIONS

**Limitation of NeMal**. NeMal is not without upper bound or limitations. The T2I model cannot generate objects that were not optimized and the hallucinations within text prompts inevitably lead to error accumulation. NeMal is constrained by the lower bound of T2I (generating consistent images with text prompts) and text prompt construction (faithful text conditions following real-world constraints). We provide some failure cases of generated images by our fine-tuned SD1.5 in Fig. 6, where the model failed to generate specified biological traits described in text prompts. We leave more discussions about the security and ethics issues of synthetic data by NeMal in Appendix.

**Conclusion**. In this work, we propose NeMal and MarineSynth, demonstrating the efficiency of synthetic data in enhancing marine visual learning. Meanwhile, NeMal formulates the first systematic and flexible framework to perform never-ending marine learning based on synthetic data, where each component within NeMal could be replaced with more powerful counterparts. We envision more powerful generative models based on more domain-specific training data and more powerful foundation models for high-quality pseudo label generation, which will lead to better synthesis and perception performance. NeMal is continuously gathering more and more marine conceptions to facilitate never-ending marine learning with minimal human efforts on both data collection and labeling.

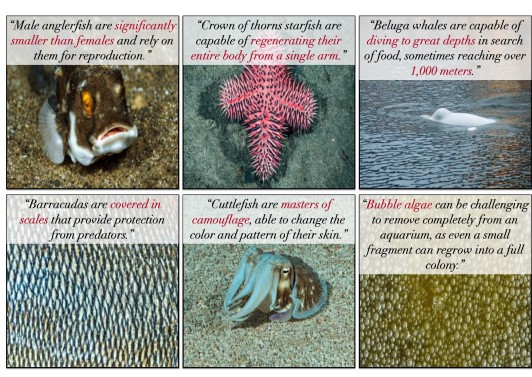

Figure 6: Failure cases of our fine-tuned SD1.5. The model failed to generate specified biological traits described in the given prompts.

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

## A APPENDIX

In this appendix, we first provide more details about formulating our internal marine image-text pair for fine-tuning the general-purpose SD1.5 model to the marine domain in Sec. A.1. We then provide more details and statistics of the constructed marine conception list and our synthetic dataset MarineSynth in Sec. A.2. We provide more implementation details of performing the tailored classification, dense segmentation, vision-language understanding and domain-specific fish counting experiments in Sec. A.3. We also compared our NeMal with few-shot learning algorithms under the same experimental setting to demonstrate its superiority. More qualitative and quantitative results and comparisons are also included in Sec. A.3. Besides, we also provide detailed and comprehensive discussions about the synthetic data regarding the backbone, security, ethics, and flexibility issues. More discussions about future directions, limitations of NeMal, and potential broader impacts of our work are provided in Sec. A.5.

### A.1 INTERNAL DATA CONSTRUCTION

To promote the marine T2I synthesis performance, we prepare 6.8 million marine text-image pairs for fine-tuning the SD1.5 model. We first scrape public Internet images from the *general marine field*. We then complement the real-world marine images for fine-tuning the general-purpose diffusion models to the marine domain, which help generate more reasonable and accurate images, and adhere to real-world data distribution while providing diverse image contents. We query the public image websites based on the following keywords:

| | | | | |
|---|---|---|---|---|
| "underwater images" | "marine images" | "underwater animals" | "tropical animal" | "island paradise" |
| "ocean paradise" | "beach" | "coastal" | "ocean animal" | "ocean waves" |
| "colorful reef creatures" | "marine mammal" | "coral reef" | "marine biodiversity" | "marine disaster" |
| "marine ecology" | "marine docks" | "deep sea diving" | "marine elements" | "marine biology" |
| "marine wastes" | "marine sustainability" | "ocean life" | "marine artist" | "marine logo" |
| "marine pattern" | "coral surveying" | "ocean science" | "marine pollution" | "fish" |
| "marine debris" | "marine organisms" | "marine engineering" | "nudibranch" | "deep sea creatures" |
| "marine science" | "marine construction" | "marine ecosystem" | "snorkeling" | "underwater ocean" |
| "marine birds" | "sea grass" | "plankton" | "oceanic abyss" | "microscopic sea life" |
| "underwater flora and fauna" | "Caribbean underwater" | "diving paradise" | "ocean movie" | "underwater life pattern" |
| "seabirds" | "aquariums" | "aquatic" | "underwater rocks" | "sea forest" |
| "marine ice" | "crustaceans" | | | |

We have scraped 5.1 million marine images from above keywords.

Then, based on our *constructed conception list* (discussed in Sec. A.2) with 2,322 marine object conceptions, we query the public image websites and download 1,000 images at most for each conception. There are very few matched marine images for some very professional marine conceptions (*e.g.*, "Laurentide ice sheet"). After filtering those repeated URLs, we have collected 1.7 million marine images with corresponding alt-texts.

Finally, we have obtained 6.8 million (5.1 million + 1.7 million) marine text-image pairs for continuously fine-tuning our text-to-image synthesis model. Through meticulously constructing the image-text pairs, we can insert the marine-specific knowledge into the text-to-image model and promote the ability to synthesize required marine images.

We fine-tune SD1.5 on our constructed marine image-text pairs. We perform the fine-tuning on 8 NVIDIA H800 GPUs with the batch size per GPU set to 48. The total fine-tuning step is 14,000 in our experiments and we will release our released model to boost the future development of marine image synthesis. With image-text pairs derived from real-world marine scenarios, our fine-tuned synthesizer could generate images with complex contexts, offering a more realistic simulation for real-world marine scenarios. Finally, we utilize the fine-tuned counterpart for marine T2I synthesis based on comprehensive text prompts. We provide more synthesized images generated from our fine-tuned SD1.5 model in Fig. 7.

We have also provided some failure cases in Fig. 8 to show the limitations of our fine-tuned SD model. As demonstrated, the diffusion model still struggles with generating very complicated and

crowded backgrounds. Furthermore, the model sometimes generates some incomplete and even wrong single objects, which do not follow real-world physical constraints. We have also observed that our fine-tuned model failed to generate clear boundaries for crowded objects. Instead, the model generated some meaningless repeated patterns. Finally, there are still some generated images with observable hallucinations.

## A.2    MARINESYNTH DATASET CONSTRUCTION

### A.2.1    CONCEPTION LIST CONSTRUCTION

To ensure the diversity, coverage, and faithfulness of synthetic data, we meticulously construct the first marine conception list, which facilitates the generation of complicated and realistic marine images, closely resembling real-world scenes and benefits domain-specific analysis. We construct our conception list from 5 different fields: 1) marine biology, 2) marine engineering, 3) marine science, 4) marine ecosystem, and 5) marine sustainability. We refer to the syllabus and the contents of marine books and official websites:

- `https://oceaninfo.com/glossary/`, which contains a large range of marine creatures.
- `https://you.stonybrook.edu/marinebio/glossary/`, a comprehensive glossary of marine biology conceptions.
- `https://www.aapa-ports.org/advocating/content.aspx?ItemNumber= 21500`, the list contains the maritime terms.
- `https://sanctuaries.noaa.gov/education/voicesofthebay/glossary. html`, the conception list contains the fishery and engineering terms.
- `https://en.wikipedia.org/wiki/Glossary_of_fishery_terms`, the glossary of fishery terms from Wiki.
- `https://www.coastalwiki.org/wiki/Definitions_of_marine_ ecological_terms`, the glossary of the marine ecological terms.
- `http://www.coml.org/investigating/glossary.html`, the glossary of the marine ecosystem and ecological terms.
- `https://texasaquaticscience.org/glossary-aquatic-water-science/ #1549336077081-22b7b9eb-a471`, the glossary of marine and aquatic science terms.
- `https://rwu.pressbooks.pub/webboceanography/back-matter/ glossary-2/`, the glossary of oceanography and engineering terms.
- `https://www.usgs.gov/glossary/ocean-glossary`, the glossary of marine geometry terms about the ocean's geologic features.
- `https://cdip.ucsd.edu/m/documents/glossary.html`, the glossary of the coastal terms and engineering terms.
- `https://worldoceanreview.com/en/glossary/`, the glossary of ocean science, ecosystem and sustainability terms.
- `https://www.sustainweb.org/goodcatch/glossary_of_seafood_terms/`, the glossary of marine sustainability terms.
- `https://www.pbs.org/emptyoceans/glossary.html`, the glossary of the ocean science and engineering terms.

After meticulously collecting the required terms from the above official websites, we ask the marine experts from the corresponding field to remove the redundant and unsatisfactory keywords/phrases. Finally, we have obtained our marine conception list with 2,322 different marine conceptions, which are comprehensive and cover the detailed marine subfields. It is important to note that our constructed marine conception list may not be exhaustive, and there could be other important and widely favored marine conceptions that have not been considered in our list. We have to admit that the marine conception list is ongoing and future exploration of the oceans will uncover additional conceptions that are not included in existing marine research. We envision a continuous revision of the marine conception list due to the dynamically changing nature of the oceans.

### A.2.2 Text Prompt Generation

Then we discuss the text prompt generation of constructing our MarineSynth dataset. There are two main sources: *Alt-texts* and *ChatGPT-generated* prompts.

**Alt-texts**. To ensure that the text captions follow the image captioning style and convey detailed information, we adopt the open-sourced LLM to rewrite these alt-texts. We adopt Mistral $8\times7B$[1] in our experiments and the instruction is

> "
> Please rewrite the following caption to make it look like the caption for one image and ensure the rewritten caption is within 20 words. "

After rewriting, we have obtained 2M text prompts for marine text-to-image synthesis.

**ChatGPT-generated**. We have also queried ChatGPT to generate diverse and comprehensive text prompts through the following instruction:

> "
> I will give you a marine term, please help generate 50 separated fact or appearance descriptions/sentences about this given marine term from different aspects (for example, biology, science, engineering, ecosystem, sustainability, culture, and others) as you can. Please make sure that we can imagine the corresponding image based on each generated description/sentence. Make sure the descriptions are within 20 words. Do not use pronouns such as "it, they, its, and their" in each generated sentence. The given marine term is ***conception***. Make sure the generated sentences follow the image caption style. Your answer should only be these 50 sentences and don't generate any other things. "

Where ***conception*** is a placeholder that comes from our carefully designed marine conception list.

Finally, we have collected 2 million text prompts from the Alt-texts and 2.3 million ChatGPT-generated text prompts. We will release all the generated text prompts to boost the future development of utilizing synthetic data for marine research.

### A.2.3 Marine Image Synthesis with Preference-based Image Picking

**Implementation details**. To promote the marine T2I synthesis performance, we construct preference-based image picking to obtain a better image with higher fidelity. We construct 100,000 one-on-one image pairs for 12 different student volunteers, who are from the marine biology field. We ask human annotators to select their preferred ones from the marine images generated from the same text prompts. The users are asked to pick up the better image, which aligns with the text prompts better. The exploration of combining human feedback in training can reduce human involvement and make the training process more efficient. By collecting feedback from users, we can guide the T2I model to synthesize images that align with the user intent. Based on collected image pairs, we have optimized a binary selector to output a choice for selecting a better one from two images. For our binary image selector, we adopt a naive ResNet-50 as the network backbone and concatenate $x_1$ and $x_2$ as the input, where $y = 0$ indicates $x_1$ is selected and vice versa. The batch size is set to 32 and we optimize the selector on our constructed 100K image pairs by 5 epochs. We adopt ResNet-18 and ResNet-50 as the network backbones to perform image classification.

Please note that we do **not** utilize such image pairs for optimizing the T2I model as Diffu-sionDPO (Wallace et al., 2023). Differently, we utilized a trained image selector for image picking at the inference stage. We repeat binary image picking by $m$ times and select the better one each time.

---

[1] https://huggingface.co/TheBloke/Mixtral-8x7B-Instruct-v0.1-GGUF

In this way, we could generate more reliable and realistic marine images since it requires multiple attempts to select an appropriate image from the SD1.5 model. Furthermore, multiple attempts to select an appropriate image based on human preferences lead to the refinement of synthesized images.

Despite being diverse and photo-realistic, the synthesized images with paired/generated pseudo labels are **task-agnostic**, which supports optimizing various sophisticated visual perception systems (classification, dense segmentation, and vision-language understanding) with pure massively synthesized data.

### A.2.4 DEBIASING

To ensure that the generated data are less biased, we have thoroughly discussed the potential biases impacting downstream tasks. We discussed the potential biases from two aspects: text prompt construction and generative models. For the text prompts, we analyzed the sources of text prompts: image captions of existing marine images, Alt-texts from the Internet, and ChatGPT-generated text prompts. We provided detailed classification results of using different text prompts, summarizing the influence of text prompts on downstream tasks from distribution alignment, faithfulness, and generalization ability. Furthermore, we construct a balanced and comprehensive marine conception list as suggested by marine experts (biologists, environmentalists, engineers, and researchers) to alleviate the potential bias. Based on the conception list, we query public marine images and balance them to obtain a more balanced distribution after fine-tuning the diffusion models. While we have made substantial progress in constructing a comprehensive and balanced conception list and ensuring the text prompts were from diverse sources, there is still a biasing problem due to human preferences when the users are querying/downloading the marine images. Considering the evolving nature of marine learning, the biasing problem is inevitable. To obtain a debiased and balanced marine data distribution, we could keep updating our marine conception list and include more diverse and efficient text prompts.

### A.3 EXPERIMENTS

NeMal proposes to harness controllable image synthesis (text conditions) for marine analysis, generating text-image data and optimizing classifiers/segmentors/VLMs based on synthetic data. In this section, we aim to demonstrate that our MarineSynth could be utilized for promoting marine visual understanding performance on **real images** even if the visual perception models are optimized from scratch or continuously fine-tuned based on **pure synthesized data**. Specifically, we perform marine image classification, coral reef segmentation, and marine vision-language understanding to demonstrate that our generated marine images are valuable for promoting various downstream marine tasks. Please note that all the models are only optimized by the synthesized images and tested by the real marine images except we especially point out. Under the marine image classification setting, we have also compared the few-shot learning algorithms when there are few-shot (*e.g.*, 5) real images available. At the end of this section, we also chose the fish counting task to better demonstrate that NeMal could effectively promote marine-specific visual understanding performance.

### A.3.1 MARINE IMAGE CLASSIFICATION

**Implementation details**. We adopt ResNet-18 and ResNet-50 as the network backbones. The batch size is set to 8 and we adopt the combination of center resizing and cropping, random flipping, and color jittering as the data augmentation to promote the robustness and generalization ability of models. The number of the training epoch is set to 20. We adopt the best model with the highest accuracy on the "IND" set to evaluate the accuracy on the other two sets.

We provide the quantitative marine image classification results under the comprehensive settings in Table 6. As illustrated, even based on pure synthetic data from the diverse text prompts, we can still achieve competitive visual recognition performance compared with the models optimized by pure real data. Meanwhile, we also notice that the models optimized by the pure synthetic data demonstrate a stronger resistance to the challenging testing ("CLG") set than the models optimized by the real data. We attribute this phenomenon to the powerful ability of our fine-tuned SD1.5 model to generate images with consistent styles as the testing set. Furthermore, incorporating real data (even 5 images) for training the classification models could achieve additional observable performance gains. Finally, due to the information leakage through the image captions, the generated images from BLIP2$^{\dagger}$ are

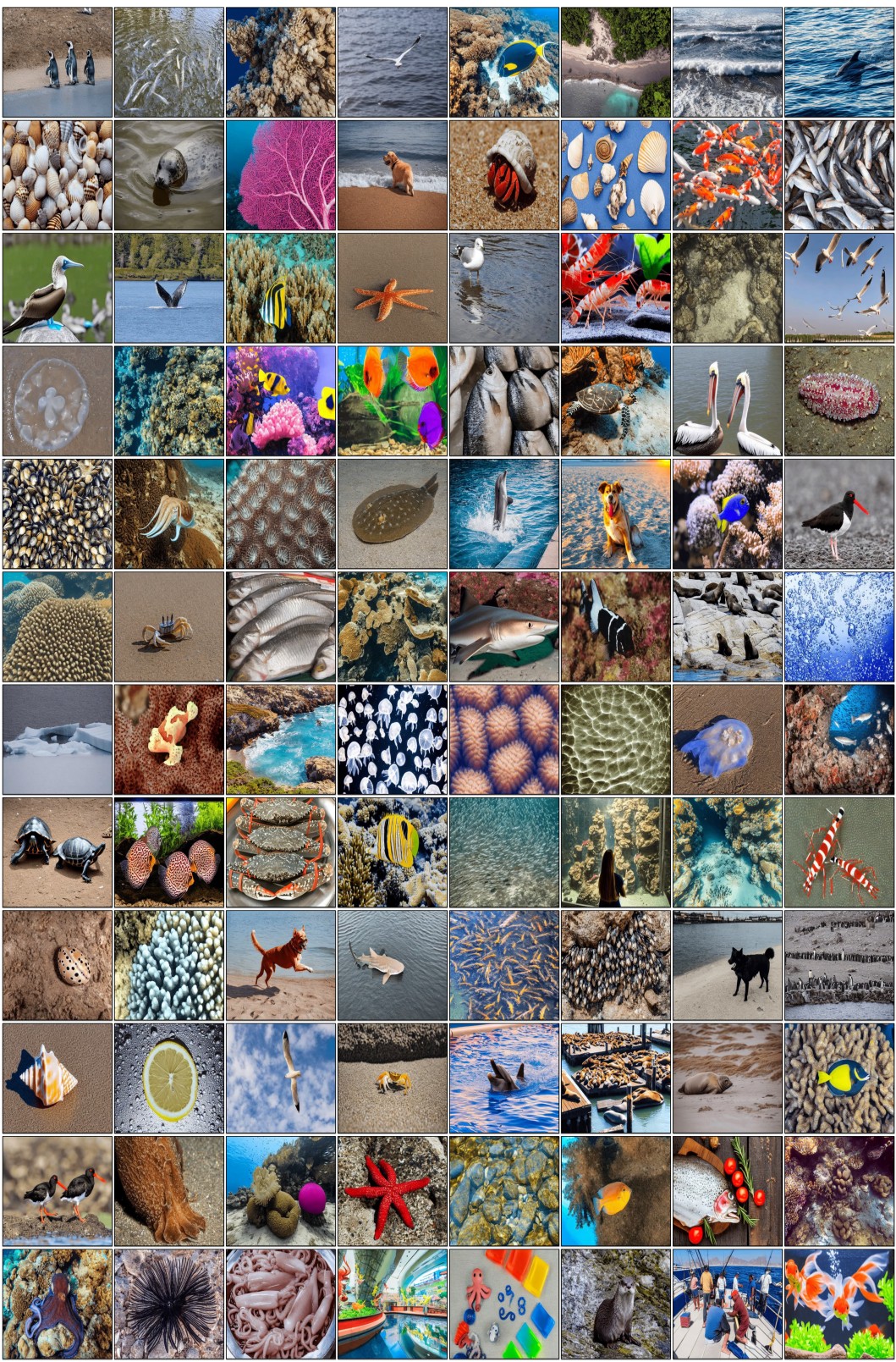

Figure 7: Example images generated by our fine-tuned SD1.5 model. Please zoom in to check more details.

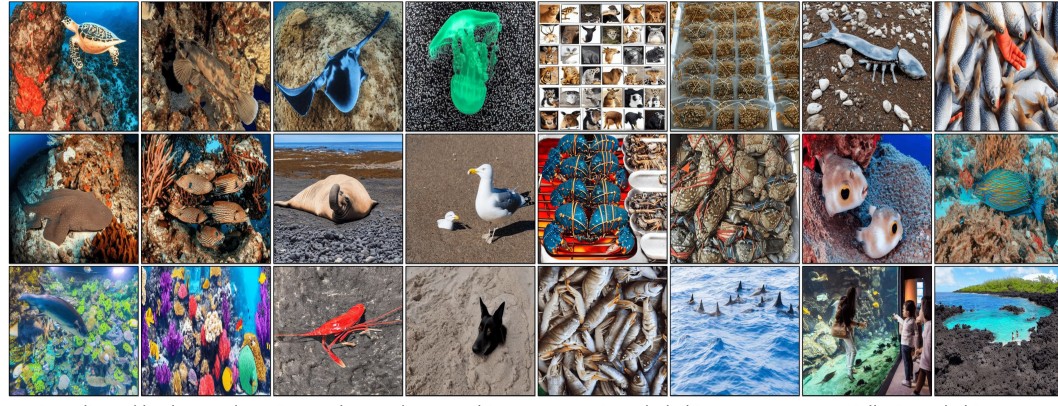

Complicated background    Incomplete and wrong objects    Crowded objects    Hallucinated objects

Figure 8: The failure cases of our fine-tuned SD1.5 on generating reasonable or complicated marine images.

limited to fixed data distribution, and adhere to available real training images, thus leading to weaker generalization ability to out-of-distribution data.

**Comparison with few-shot learning algorithms**. We also consider the experimental setting, where there are few-shot real data exist. Under this setting, there are two lines of works, which could also augment the limited real-world data: domain adaptation (Zhu et al., 2017) and few-shot learning (FSL) algorithms (Snell et al., 2017; Sung et al., 2018). We first explain some essential differences between NeMal and domain adaptation or few-shot learning (FSL) techniques.

- Domain adaptation techniques were designed to handle domain shift problems where there are clear image domains: appearances shift while conceptions are consistent (*e.g.*, water turbidity *i.e.* green-like vs blue-like water).
- FSL algorithms hold when few-shot examples are given for some novel object categories.

However, both domain adaptation and few-shot learning techniques cannot handle the out-of-distribution/novel object conceptions without real images and these methods must have access to real images. In contrast, given object conceptions/categories, NeMal could generate text prompts and synthesize corresponding marine images to optimize models. Furthermore, NeMal could utilize the power of large-scale pre-training to generate required images for optimizing the models in a never-ending learning manner.

Here, we compare NeMal with two representative few-shot learning algorithms (Prototypical Network (Snell et al., 2017) and Relation Network (Sung et al., 2018)). We skip domain adaptation algorithms as we found that obtaining clear image domain definitions for domain adaptation in our case is not practical. Following the problem formulation of FSL, we adopt the ImageNet-100 as the training set and the training split of Sea Animal dataset as the support set. The categories are disjoint between the training set and the support set. The constructed IND, OOD and CLG sets are regarded as testing set. To make a fair comparison, we perform experiments under the 23-way 5-shot setting. During the evaluation stage, 600 episodes (15 random query images per class) are randomly constructed from the testing set to obtain the results. Our NeMal is stronger than the FSL algorithms by synthesizing marine images from the required marine object categories.

### A.3.2    CORAL REEF SEGMENTATION

**Implementation details**. We synthesize 20,000 coral reef images based on corresponding rewritten text prompts from the Alt-texts. Then we utilize these diverse synthesized coral reef images for optimizing models for coral reef segmentation. We infer the CoralSCOP (Zheng et al., 2024) (with the ViT-B backbone) with these synthesized coral reef images for coral mask generation: automatic setting with $32 \times 32$ grid points as prompts. We set the IoU threshold to 0.82 to filter out those low-quality coral masks. Then we regard the generated coral masks after filtering as the pseudo labels for optimizing various dense segmentation algorithms. To provide a better illustration, we provide the synthesized coral reef images with the generated coral mask (pseudo labels) in Fig. 9. For

Table 6: Quantitative classification results (Top-1 accuracy, higher is better) of different models optimized under various settings. Avg. indicates the average accuracy of the "IND", "OOD" and "CLG" sets. † indicates that BLIP2 must require real images for image caption generation as text prompts. **Note all the testing images are real images**.

| Methods | ResNet-18 | | | | ResNet-50 | | | |
|---|---|---|---|---|---|---|---|---|
| | IND | OOD | CLG | Avg. | IND | OOD | CLG | Avg. |
| Oracle (pure real data) | 74.65 | 56.91 | 36.29 | 55.95 | 75.82 | 59.30 | 38.37 | 57.83 |
| *Pure synthetic data* | | | | | | | | |
| ChatGPT+Alt-texts | 49.26 | 59.87 | 45.12 | 51.42 | 53.11 | 61.48 | 46.38 | 53.66 |
| ChatGPT | 46.48 | 54.96 | 35.84 | 45.76 | 52.98 | 59.48 | 41.97 | 51.48 |
| Alt-texts | 43.87 | 57.43 | 42.37 | 47.89 | 48.06 | 57.30 | 47.87 | 51.08 |
| BLIP2† | 53.43 | 51.22 | 34.84 | 46.50 | 54.37 | 49.91 | 32.65 | 45.64 |
| *Pure synthetic data + Oracle* | | | | | | | | |
| ChatGPT + Alt-texts + Oracle | 75.43 | **70.83** | **49.17** | **65.14** | 77.30 | 70.52 | **53.58** | **67.13** |
| ChatGPT + Oracle | 74.61 | 70.35 | 47.16 | 64.04 | **77.65** | **72.30** | 50.80 | 66.92 |
| Alt-texts + Oracle | 74.87 | 68.48 | 45.64 | 63.00 | 76.78 | 69.61 | 46.27 | 64.22 |
| BLIP2† + Oracle | **75.83** | 62.22 | 40.15 | 59.40 | 76.87 | 66.17 | 43.19 | 62.08 |
| *Few-shot real data* | | | | | | | | |
| 5-shot | 34.52 | 23.48 | 14.73 | 24.24 | 35.62 | 25.13 | 16.10 | 25.62 |
| 10-shot | 42.74 | 30.91 | 19.37 | 31.01 | 44.89 | 30.14 | 18.85 | 31.29 |
| 20-shot | 49.70 | 38.22 | 22.23 | 36.72 | 53.20 | 39.04 | 23.41 | 38.55 |
| 50-shot | 60.87 | 43.22 | 28.50 | 44.20 | 63.38 | 44.70 | 27.83 | 45.30 |
| 100-shot | 66.70 | 46.78 | 29.24 | 47.57 | 69.20 | 54.39 | 33.32 | 52.30 |
| Imbalanced | 35.57 | 27.13 | 16.99 | 26.56 | 37.80 | 28.74 | 19.11 | 28.55 |
| *Few-shot real data + synthetic data* | | | | | | | | |
| 5-shot + ChatGPT + Alt-texts | 56.66 | 58.7 | 44.94 | 53.43 | 58.98 | 65.17 | 47.61 | 57.25 |
| 5-shot + ChatGPT | 52.43 | 55.74 | 38.48 | 48.88 | 56.33 | 57.74 | 39.00 | 51.02 |
| 5-shot + Alt-texts | 57.04 | 61.91 | 44.12 | 54.36 | 59.24 | 61.61 | 47.83 | 56.23 |
| 5-shot + BLIP2† | 56.13 | 50.87 | 33.36 | 46.79 | 58.03 | 51.09 | 34.77 | 47.96 |
| 10-shot + ChatGPT + Alt-texts | 60.03 | 62.04 | 43.45 | 55.17 | 62.77 | 62.61 | 47.76 | 57.71 |
| 10-shot + ChatGPT | 56.91 | 57.61 | 37.44 | 50.65 | 60.37 | 58.22 | 39.33 | 52.64 |
| 10-shot + Alt-texts | 56.74 | 57.39 | 42.89 | 52.34 | 61.11 | 63.65 | 47.83 | 57.53 |
| 10-shot + BLIP2† | 59.39 | 51.00 | 34.77 | 48.39 | 62.46 | 54.09 | 34.32 | 50.29 |
| 20-shot + ChatGPT + Alt-texts | 62.16 | 63.30 | 42.97 | 56.14 | 66.46 | 69.26 | 47.72 | 61.15 |
| 20-shot + ChatGPT | 59.63 | 59.57 | 41.22 | 53.47 | 63.16 | 62.13 | 42.30 | 55.86 |
| 20-shot + Alt-texts | 59.65 | 62.65 | 44.42 | 55.57 | 61.98 | 63.83 | 47.20 | 57.67 |
| 20-shot + BLIP2† | 63.39 | 53.09 | 33.43 | 49.97 | 64.64 | 57.91 | 37.22 | 53.26 |
| 50-shot + ChatGPT + Alt-texts | 68.30 | 66.26 | 46.64 | 60.40 | 70.99 | 67.48 | 48.12 | 62.20 |
| 50-shot + ChatGPT | 66.74 | 62.57 | 38.96 | 56.09 | 69.84 | 65.83 | 45.12 | 60.26 |
| 50-shot + Alt-texts | 65.91 | 63.74 | 44.23 | 57.96 | 67.86 | 64.13 | 47.09 | 59.69 |
| 50-shot + BLIP2† | 68.57 | 56.65 | 36.59 | 53.94 | 70.20 | 60.09 | 38.03 | 56.11 |
| 100-shot + ChatGPT + Alt-texts | 71.09 | 68.52 | 49.13 | 62.91 | 74.38 | 70.09 | 49.53 | 64.67 |
| 100-shot + ChatGPT | 70.26 | 65.22 | 42.63 | 59.37 | 72.16 | 65.13 | 42.86 | 60.05 |
| 100-shot + Alt-texts | 70.61 | 66.22 | 44.71 | 60.51 | 71.73 | 68.34 | 47.61 | 62.56 |
| 100-shot + BLIP2† | 71.13 | 61.87 | 39.26 | 57.42 | 73.77 | 63.17 | 40.48 | 59.14 |
| Imbalanced + ChatGPT + Alt-texts | 56.30 | 62.70 | 45.68 | 54.89 | 60.46 | 61.83 | 42.15 | 54.81 |
| Imbalanced + ChatGPT | 53.74 | 55.78 | 39.55 | 49.69 | 55.72 | 57.74 | 38.81 | 50.76 |
| Imbalanced + Alt-texts | 50.48 | 53.22 | 38.44 | 47.38 | 54.28 | 58.09 | 43.82 | 52.06 |
| Imbalanced + BLIP2† | 53.48 | 49.96 | 34.03 | 45.82 | 55.98 | 50.65 | 36.07 | 47.57 |

DeepLabV3 (Chen et al., 2017) and SegFormer (Xie et al., 2021), we optimize the models following the official instructions and set the number of total iterations to 80,000. When fine-tuning SAM to perform coral reef segmentation, we freeze the image encoder and only optimize the mask decoder and the prompt encoder. The training prompt is only the point prompt, where we adopt three random points inside the generated coral mask as prompt following the training recipe of SAM (Kirillov et al., 2023). Please note we do not adopt the bounding box prompt during the training procedure for

Table 7: Quantitative classification results (Top-1 accuracy) of different algorithms on Sea Animal dataset.

| Settings | IND | OOD | CLG | Avg. |
|---|---|---|---|---|
| 5-shot | 35.62 | 25.13 | 16.10 | 25.62 |
| 5-shot + Prototypical Network | 43.45 | 35.76 | 21.43 | 33.55 |
| 5-shot + Relation Network | 45.21 | 36.51 | 23.71 | 35.14 |
| 5-shot + NeMal | **56.33** | **57.74** | **39.00** | **51.02** |

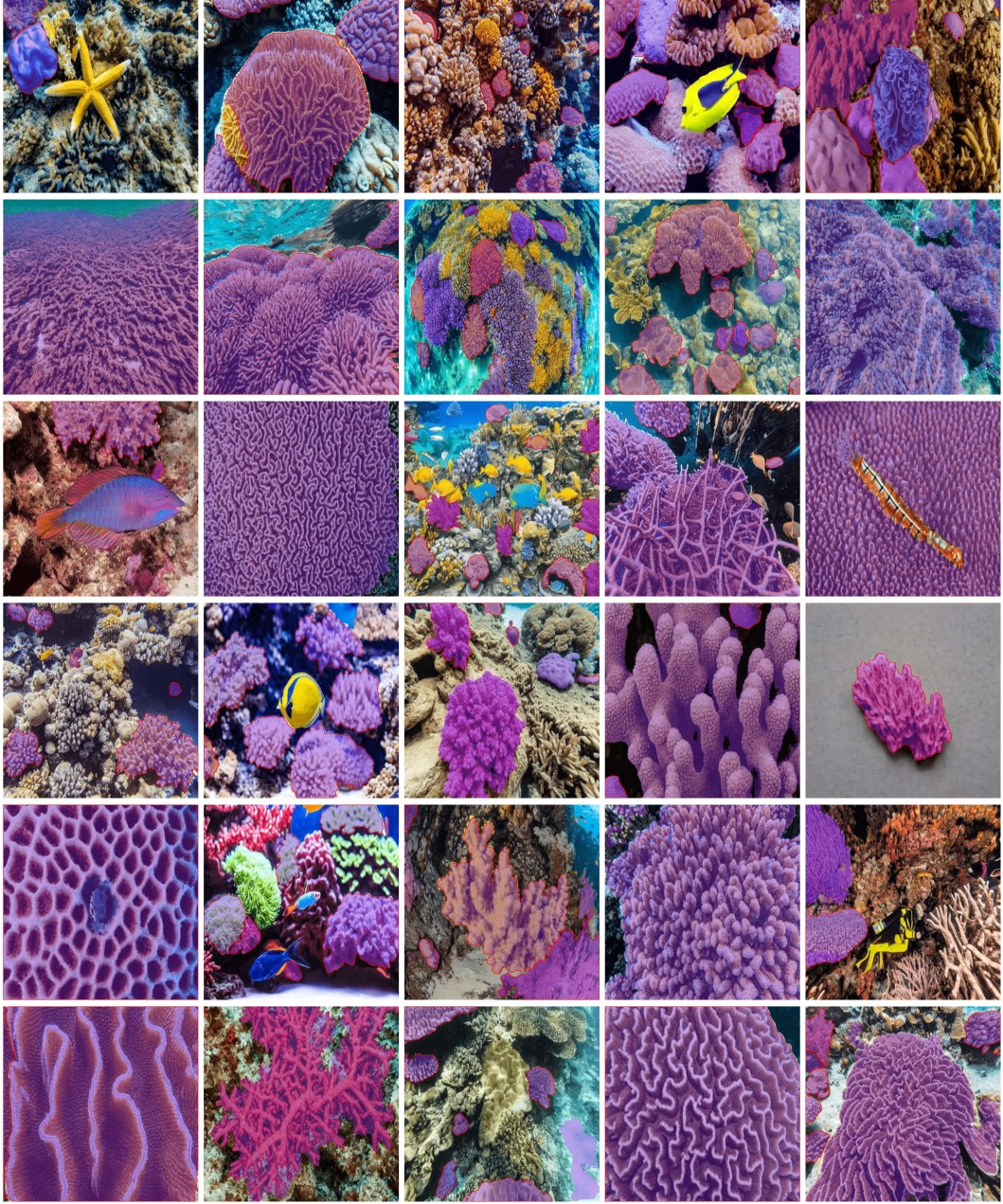

Figure 9: The visualization of synthesized coral reef images with pseudo coral mask labels (highlighted in purple) from the foundation model. Best viewed in color.

Table 8: The object counting performance under two settings.

| Methods | MSE↓ | MAE↓ | NAE↓ |
|---|---|---|---|
| CLTR | 17.47 | 37.06 | 0.29 |
| CLTR + NeMal | **17.01** | **36.43** | **0.27** |

fine-tuning SAM. We set the number of the total training steps to 100,000 and the batch size per GPU to 1. We perform the fine-tuning experiments on 4 RTX 3090 GPUs.

Please note at the testing stage, we adopt 400 real coral reef images with ground truth labeled by coral biologists for testing. We evaluate SAM and the fine-tuned counterpart under the 1) "automatic" setting where the $32 \times 32$ grid points are utilized as point prompts for generating coral reef masks and 2) "point prompt" setting where one random point inside each coral mask (ground truth) is used as prompt for coral mask generation.

### A.3.3 MARINE VISION-LANGUAGE UNDERSTANDING

**Testing set construction**. We first provide examples of our constructed testing sets to evaluate the vision-language understanding performance of various VLMs in Fig. 10. To quantitatively evaluate the performance of VLMs on marine image comprehension, we adopt the binary answers to better evaluate the ability of the models. The constructed testing sets cover questions about biological identification, appearance description, object counting, event detection, detailed reasoning, common sense query, motivation explanation, scientific reading, logo recognition, and abstract image understanding.

**Implementation details**. We perform the fine-tuning on our constructed MarineSynth dataset. The synthesized images are paired with the provided text prompts as the text captions. We follow the official settings of MiniGPT4 (Zhu et al., 2023), LLaVa (Liu et al., 2023b) and LLaVa-1.5 (Liu et al., 2023a) to promote the performance of the marine image understanding. With the significant scale of readily available image-text pair (text prompts and corresponding synthesized marine image), we could further boost the vision-language understanding performance based on various VLMs.

### A.3.4 FISH COUNTING

We have also explored an interesting fish counting task to perform an experiment on the IOCFish5K dataset (Sun et al., 2023). Due to the IOCFormer (proposed in IOCFish5K dataset (Sun et al., 2023)) was not open-sourced, we adopted CLTR (Liang et al., 2022) as our baseline to perform experiments. First, we follow the official dataset split of the IOCFish5K dataset to optimize CLTR and adopt the evaluation metrics (MSE, MAE and NAE) to measure the counting performance. Then we pick up 1,000 marine images with crowded marine objects (*e.g.*, fish, crab, shell, sea lion, and so on) from our MarineSynth dataset by human labelers. We utilize the optimized CLTR model to generate pseudo labels. Finally, we combine the original real images with GTs and the synthesized marine images with generated pseudo labels for continuous training. We report the result comparison on the validation set of the IOCFish5K dataset as follows:

As can be seen in Table 8, NeMal's synthetic images and pseudo labels can improve object counting performance over all metrics. Compared with existing marine datasets like IOCFish5K, NeMal only requires a few human efforts (*e.g.*, text prompt generation and image picking) to synthesize marine images. Since NeMal could synthesize task-agnostic marine images, we could combine various domain-specific requirements into prompt design and pseudo label generation for corresponding marine applications.

### A.4 DISSECTING SYNTHETIC DATA

In this section, we first explore the effectiveness of utilizing the synthesized data from different diffusion models. Then we comprehensively discuss the security and ethics issues of the synthesized data. Finally, we evaluate the performance of NeMal on synthesizing some endangered marine species to demonstrate its feasibility.

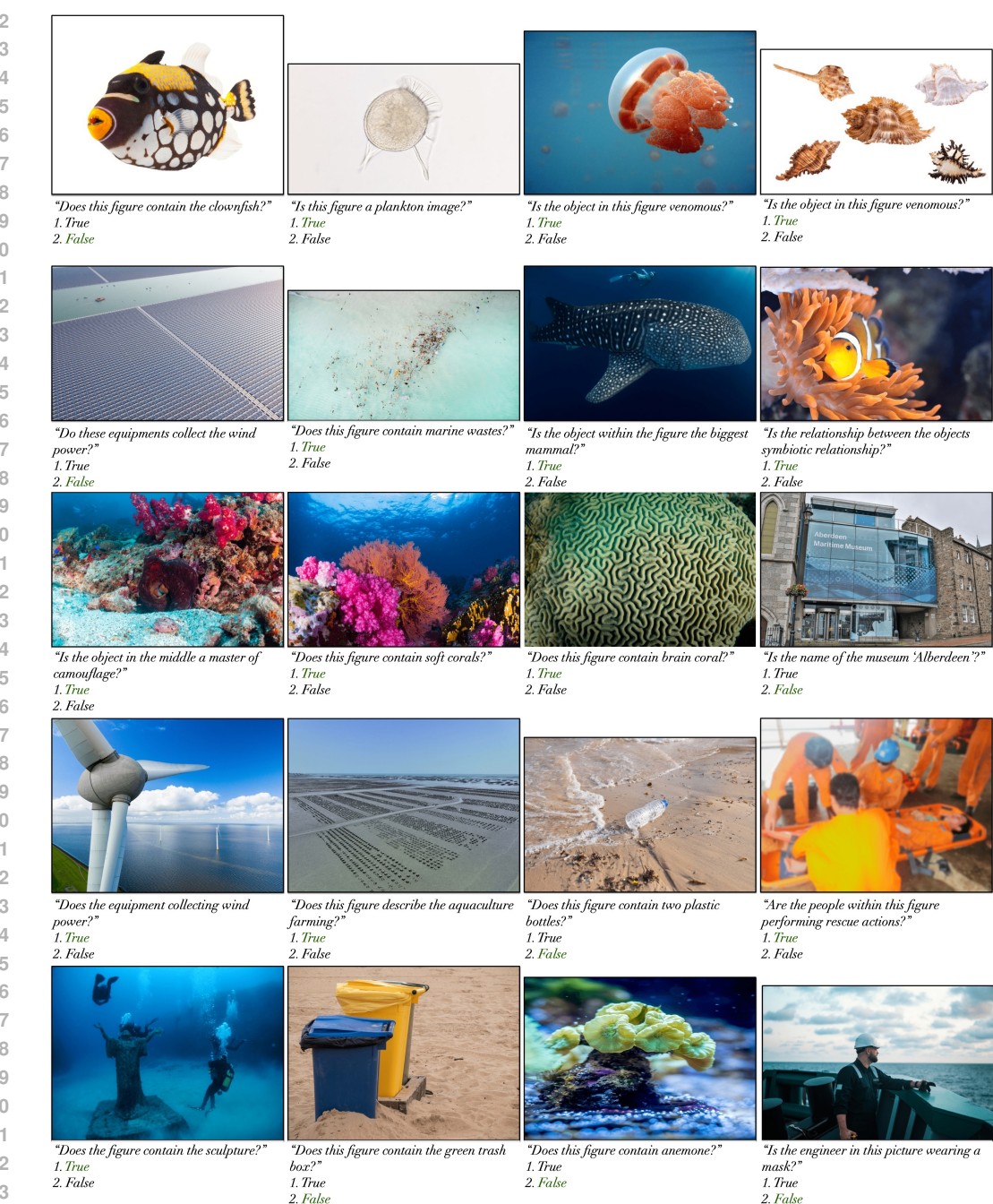

Figure 10: Examples of constructed testing sets from different aspects for evaluating marine vision-language understanding. We adopt the binary answers to better measure the ability of VLMs to accurately understand the marine images.

### A.4.1 COMPARING VARIOUS DIFFUSION MODELS

Besides SD1.5, we further compare NeMal with two recent diffusion models SDXL (Podell et al., 2023) and SD2.1 (Rombach et al., 2022a). Following the same experimental setting, we adopt the same text prompts and generate corresponding marine images based on the provided official models. We adopt the same 500 ChatGPT-generated text prompts per category and the network backbone is ResNet-50. Then we optimize the classifiers based on the synthetic images and report the classification accuracy of the trained models as follows: After the fine-tuning on collected marine

Table 9: Quantitative classification results (Top-1 accuracy) of models optimized by synthetic images generated by different diffusion models.

| Settings | IND | OOD | CLG | Avg. |
|---|---|---|---|---|
| SD1.5 | 42.74 | 54.87 | 39.74 | 45.78 |
| SDXL | 41.87 | 54.30 | 40.22 | 45.46 |
| SD2.1 | 44.56 | 55.39 | 38.90 | 46.28 |
| Fine-tuned SD1.5 (NeMal) | **52.98** | **59.48** | **41.97** | **51.48** |

data, NeMal shares a stronger ability to synthesize required and more accurate marine images, which could further promote the marine visual perception performance of the downstream tasks as illustrated in Table 9. We provide more qualitative comparisons in Fig. 11.

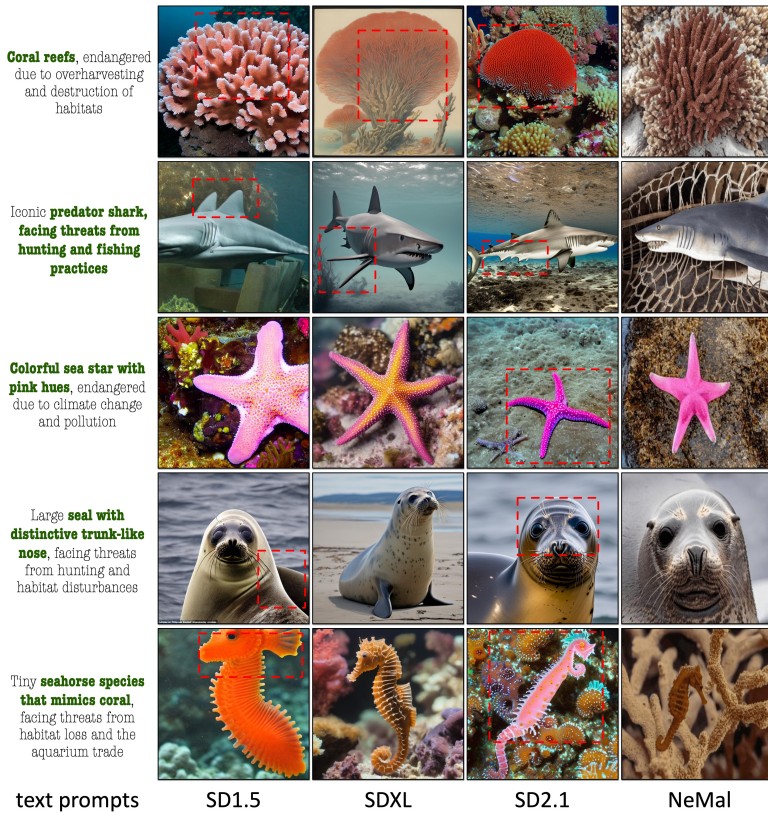

Figure 11: Qualitative comparisons between SD1.5, SDXL, SD2.1 and NeMal (fine-tuned SD1.5 based on our collected marine data). The regions covered by red boxes indicate wrongly generated parts, which did not follow physical conditions. Our NeMal could generate more faithful images that adhere to real-world marine images.

### A.4.2 SECURITY AND ETHICS

In this section, we discuss and evaluate the security and ethics issues of the synthesized data from the diffusion models. We first emphasize that the generated marine images are the **intermediate by-products** of our system as demonstrated in Fig. 1 of our main paper. Our goal is to utilize the synthetic data from SD models with generated pseudo labels from text prompts or foundation models to promote the perception performance of various downstream tasks. The synthetic data could be **immediately destroyed** after the training of the visual perception models. All the testing data used in our paper are real data to report the possibility of utilizing synthetic data from SD models for optimizing models that could be used for evaluating real data. The main focus of this work is

the visual perception models, which yield high-level semantics rather than generating novel image contents.

We then list the efforts we made in our work to avoid the potential security issue:

- During the data generation process, the text prompts are from our constructed conception list, which is based on existing marine glossaries. The glossaries have already been checked by some official institutions such as NOAA to avoid potential security issues. Meanwhile, we have checked the prompts used for image generation, they are all related to generating marine creatures or scenarios.

- Furthermore, we have collected public marine images from some official websites (*e.g.*, EOL, Flickr, Shutterstock) for fine-tuning the StableDiffusion model. The marine images have been pre-screened, and released by these authoritative institutions so that it is less likely to have harmful images. By fine-tuning the SD model on these marine images, we could help the SD model generate appropriate images.

- Human preference has been combined into the data generation procedure to reduce the risk of generating unsafe images.

We also perform large-scale safety testing to evaluate the security of the synthesized data. We conducted a larger-scale calculation of the inappropriate content (Schramowski et al., 2022) ratio by randomly sampling 40,000 images and the inappropriate ratio is **0.0331**. The inappropriate ratio of real images of the authoritative institution Shutterstock is **0.0341**. The results indicate that our generated data is in a similar level of as publicly available real marine images.

As for the evaluation metric, the inappropriate probability is to evaluate the inappropriate/unsafe contents generated from generated models and it has been widely used by previous notable visual safety-related research works (Schramowski et al., 2023; Gandikota et al., 2023; Kumari et al., 2023; Lee et al., 2024). According to (Schramowski et al., 2022), inappropriate content is defined as *defamatory*, *false*, *inaccurate*, *abusive*, *indecent*, *obscene* or *menacing*, or otherwise *offensive*. Our results indicate that the proportion of inappropriate content in the synthetic dataset is comparable to the real marine images: 0.0331 (synthetic data) vs 0.0341 (real data), or even lower than, that in many widely used datasets such as MS-COCO: 0.0331 (ours) vs 0.058 (MS-COCO).

Finally, while all generative models face data security issues, many models like ChatGPT (OpenAI, 2022), Midjourney, and StableDiffusion (Rombach et al., 2022a) are already in widespread use and provide online services. Effective harmful content processing can address this concern. We have implemented better safety measures during data collection and will incorporate the inappropriate detection methods to process the generated data. Finally, for our MarineSynth4M dataset, we will perform human checking with the help of inappropriate detection methods to try our best to avoid the potential security issue of synthetic marine images. We will responsibly release the synthetic data and include specific licensing terms, requiring applications from researchers for research purposes only, similar to the application process for downloading the model weights like LLaMA (Touvron et al., 2023).

### A.4.3 SYNTHESIZING ENDANGERED MARINE SPECIES

We believe that it is an interesting use case to synthesize some endangered marine species, and therefore provide the qualitative results of NeMal on synthesizing rare or endangered marine species in Fig. 12. The rare marine species' names were listed by ChatGPT. NeMal could generate reasonable outputs for some rare classes due to our constructed balanced conception list and collected marine data for domain-specific fine-tuning.

## A.5 DISCUSSIONS

### A.5.1 CONTRIBUTION CLAIM

The main contribution of this paper is to propose the first never-ending marine learning framework by combining controllable image synthesis and powerful foundation models to reduce human efforts on both data collection and labeling. Even the individual components of our NeMal are inherited from existing works, combining them together to build a never-ending system is not a trivial task. We provide insights on how to design and build a never-ending system and demonstrate the promise of

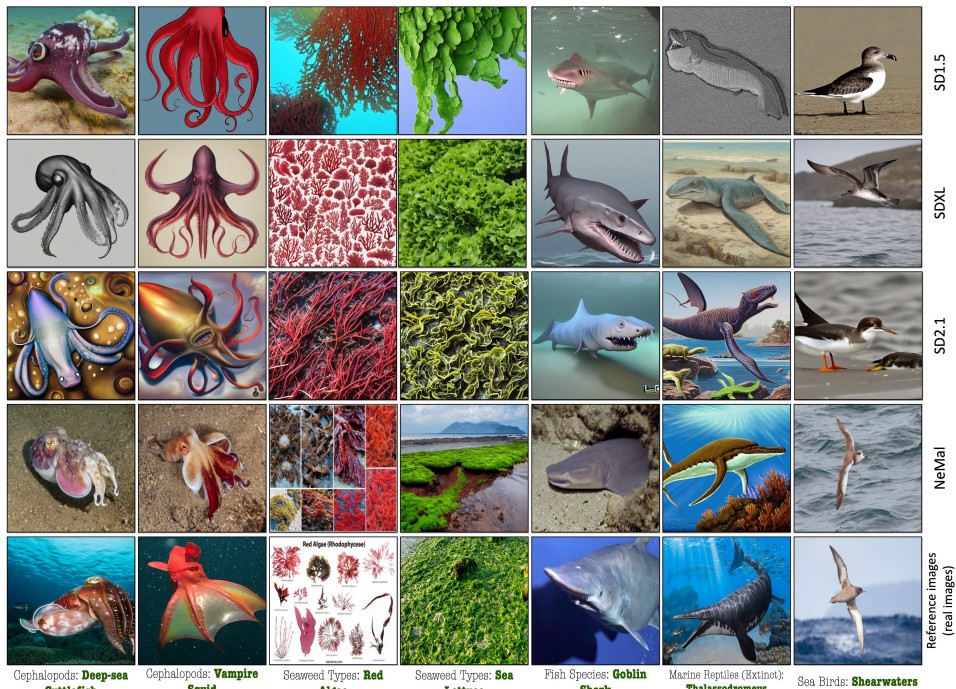

Figure 12: Qualitative results of SD1.5, SDXL, SD2.1 and NeMal on generating some rare/uncommon or endangered marine species. The species names were generated by ChatGPT and the real reference images (last row) are included for better comparison.

utilizing synthetic data for scientific research. The domain-specific expertise is combined to make NeMal more efficient. The proposed framework and method could be also extended to other fields and applications for reducing the human efforts on both data collection and labeling.

We have performed comprehensive and hierarchical analysis and ablation studies on each component of the whole framework: 1) how to generate reasonable text prompts 2) synthesize more high-fidelity and aligned images by combining human preferences 3) explore the effectiveness of domain-specific fine-tuning, and 4) the gap between synthesized images and real images. Our experiments and observations provided insights on how to utilize synthetic data for domain-specific research and how to design a never-ending system. The fundamental problem in this work is try to answer whether we could utilize the synthetic data for marine visual research. To answer this fundamental problem, we have performed various experiments to demonstrate the promise of utilizing synthetic data to promote perception performance. We emphasize again that addressing the widespread security issues of generative models is not our main focus. Our main focus of this work is to promote the marine visual perception performance based on synthetic data or a mix of synthetic data and real data. The synthetic data is only the intermediate by-products of our framework and all the testing data is from the real world.

**Comparison with existing works**. We also noticed a recent work (Shumailov et al., 2024) concluded that utilizing the synthetic data for optimizing models recursively will lead to model collapse. However, the direct transfer of the conclusion of (Shumailov et al., 2024) to our work lacks supportive evidence. The phenomenon concluded in (Shumailov et al., 2024) was based on VAE architecture (Kingma, 2013) and an ideal condition. The setting in (Shumailov et al., 2024) is different from our setting. The conclusion based on GMM (Bond, 2001) or VAE cannot be transferred to GPT (OpenAI, 2022) or Diffusion models (Rombach et al., 2022a) directly. Problem formulation is different between (Shumailov et al., 2024) and our work. Our main focus of this work is to utilize synthetic data for promoting various perception models (classification, segmentation and VLMs). (Shumailov et al., 2024) was analyzed under the setting that the synthetic data (output of the generative models) were used to optimize the generative models themselves, repeatedly. Under the ideal conditions, no corrections were included. However, our work is to use synthetic data to optimize visual perception models. **Please note the synthetic data is the output of generative models while**

**the synthetic data is the input of the perception models (output is high-level semantics and task-dependent)**. The generative models are optimized in pixel space while perception models are optimized in semantic space. It is believed that the latter perception models are easier to optimize and thus have fewer constraints on the training data. The generated pseudo labels from other foundation models also inserted additional/external knowledge into the whole learning process. Thus, the settings or problem formulation in (Shumailov et al., 2024) and our work are very different.

### A.5.2 UPPER BOUND OF NEMAL

In theory, it is inevitable to have a performance improvement upper bound even if we continuously synthesize marine images or if we only use pure real data. Such improvement may diminish due to various factors, such as network compactness, data noise, data diversity, and distribution shift. Empirically, this upper bound would continue to expand as our framework benefits from more powerful foundation models and more efficient data selection strategies.

There are various factors that affect the quality of the generated data: 1) the alignment between the generated images and text prompts; 2) the image quality of the synthesized images; 3) the distribution shift between the testing data and generated training data. We have made the following attempts to handle these factors:

- For the alignment between the synthesized images and the text prompts, we have included various sources of text prompts to promote the diversity, coverage and faithfulness of generated images.

- To ensure the image quality of the synthesized images, we collect 6.8M marine images based on our carefully constructed marine conception list for domain-specific fine-tuning. We demonstrated that fine-tuning could promote the quality of synthesized marine images in Table 3 (higher classification accuracy indicates better image quality). We combined the human preference to do preference-based image picking to further promote the image quality.

- For the distribution shift problem, we could constantly synthesize marine images. However, due to the image quality and training data noise issues, we cannot constantly promote classification accuracy. Please also note that human-constructed testing data is also a subset of real-world endless marine data in a never-ending manner. With the intrinsic distribution shift between the training data and testing data, in theory, we cannot constantly promote perception accuracy by continuously synthesizing more training data even our synthesized data are 100% accurate.

### A.5.3 EXTERNAL KNOWLEDGE INJECTION

Even though we have tried our best to construct a comprehensive and balanced marine conception list, the marine conception list is ongoing due to the dynamic changing nature of the oceans. The researchers are continuously proposing novel marine conceptions and the marine sciences are keeping evolving. To continuously learn marine knowledge and perform never-ending marine learning, we plan to insert external knowledge into our built system. We aim to combine new instances/conceptions from the unseen training corpus with the existing foundation models without re-training the foundation models to reduce the fine-tuning efforts. Our goal is to expand the language-vision dictionary of the foundation models through an external knowledge bank so that it can bind new conceptions of what the users want to generate or recognize with learned knowledge within the models. Inspired by MyVLM (Alaluf et al., 2024), we plan to design the external memory bank to store the inserted knowledge while utilizing the strong semantic priors learned from a significant collection of image-text pairs. We leave this as our future work.

### A.5.4 COMBINING TASK-SPECIFIC FEEDBACK

In this work, we proposed to adopt the marine T2I as the surrogate to synthesize corresponding images with corresponding pseudo labels for enhancing marine visual understanding. The task-specific feedback could also be utilized to further promote the data quality of synthesized marine images thus leading to better recognition performance. We take the image classification as the illustration example. The entropy distribution of the real testing images on the optimized classification models could reveal the challenging categories. With the feedback from the specific visual recognition task, we can perform better marine image synthesis, formulating a mutually beneficial cycle. Furthermore, continuous feedback can also be used to modify generation parameters for better synthesis performances.

However, we still need to avoid error accumulation within the cycle since it is likely to produce some noisy or fully wrong outputs/labels, which may lead to degraded synthesis and recognition performance. Meanwhile, how to design a reasonable and appropriate condition to terminate the mutual cycle is still a challenging problem.

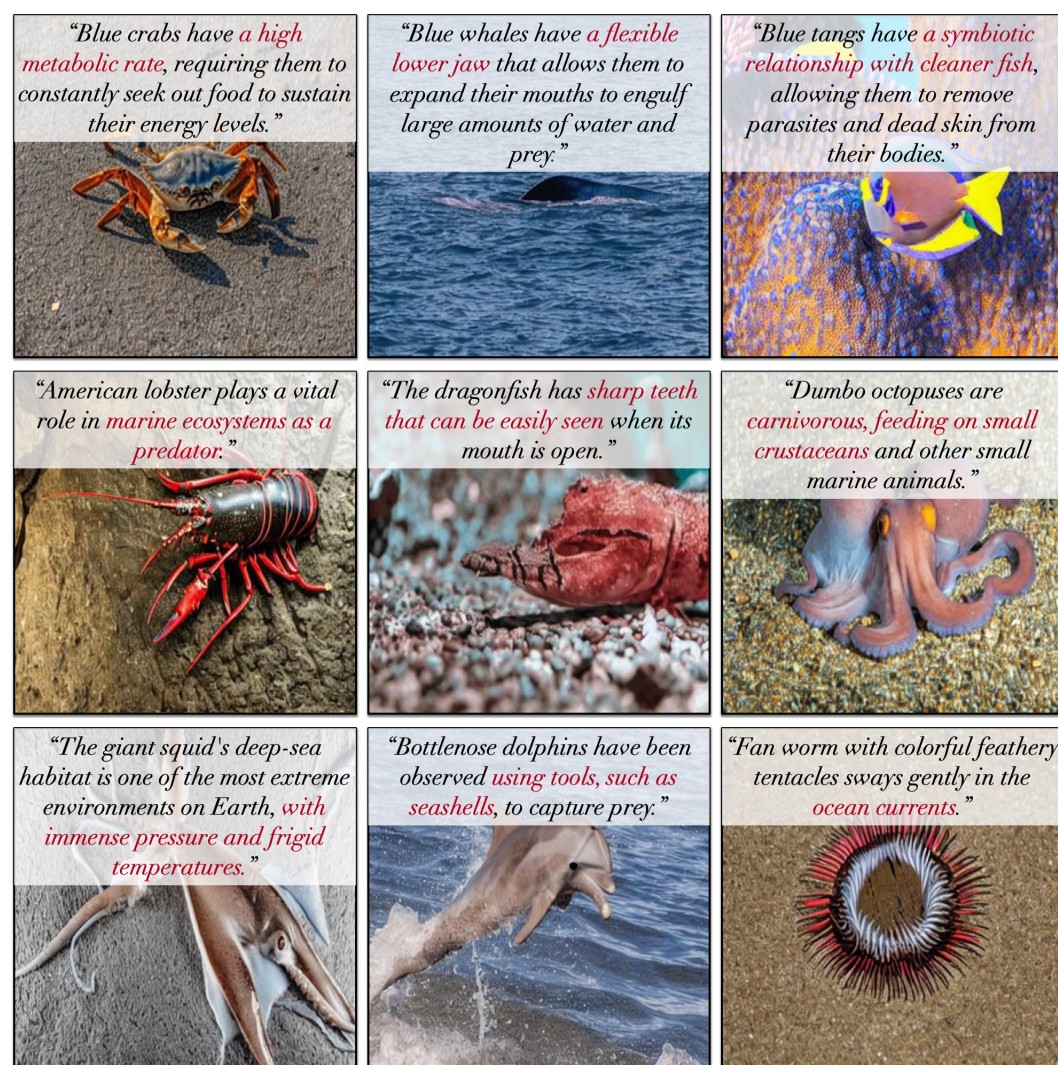

Figure 13: The text-image misalignment between the generated images and the provided text prompts. Best viewed in color.

### A.5.5 LIMITATIONS

Our NeMal is not without limitations. We discuss the following limitations of our NeMal as follows:

**Biasing problem**. While we have made substantial progress in constructing a comprehensive and balanced conception list and ensuring the text prompts were from diverse sources, there is still a biasing problem due to human preferences when the users are querying/downloading the marine images.

**Coverage**. Our current marine conception list may not be exhaustive. For instance, we only focus on marine biology, ecosystem, engineering, science, and sustainability. There may be other factors that warrant consideration. The marine conception list should be ongoing and updated based on the domain requirements.

**Text-image misalignment**. The current marine text-to-image synthesis is still far from satisfying the requirement for generating meaningful and reasonable scientific images for domain-specific research.

As demonstrated in Fig. 13, the diffusion models cannot generate the required scientific figures even after being fine-tuned to the marine domain. In our future work, we intend to collect figure-caption pairs from scientific books and articles to enable a more reasonable image synthesis.

**Hallucinations**. We acknowledge the mild hallucinations or noises within the generated pseudo labels from generated text prompts (with intrinsic hallucinations) or foundation models. Mitigating hallucinations or noises has been an ongoing research for LLMs and image diffusion models. In our case, NeMal would benefit from hallucination detection, confidence-based label selection, or pseudo label refinement to yield more accurate supervision.

### A.5.6 BROADER IMPACTS

In this work, we have demonstrated the promise of utilizing synthetic data for performing domain-specific/marine research. We have proposed a systematic framework on how to assemble the generation of meaningful text prompts, power T2I synthesis model, foundation models for pseudo label generation, and model optimization based on task-specific supervision. The insights on how to design a never-ending system: fine-tuning general-purpose SD to domain-specific counterpart, domain conception construction, text prompt generation, and preference-based image picking could be extended to other domains as well. Our NeMal is also the first attempt to propose the never-ending marine learning system, which embraces a philosophy of ceaseless learning in marine science. NeMal is essential because the marine environment is in a constant state of flux, with novel conceptions, ecological relationships, and biological features awaiting discovery.

