# OpenReview forum: "NeMal: Never ending Marine Learning - Unleashing the Power of Controllable Image Synthesis for Promoting Marine Visual Understanding"
_ICLR.cc/2025/Conference — Submitted to ICLR 2025_

### Official Review · Reviewer_g6bw · 2024-10-30

**Soundness:** 3
**Presentation:** 3
**Contribution:** 3
**Rating:** 5
**Confidence:** 4

**Summary:**

This work focused on marine visual learning. It proposed systematic and flexible framework to perform never-ending marine learning based on synthetic data, called NeMal. Experiments proved the effectiveness of the proposed pipeline and data.

**Strengths:**

1. This work proposed a pipeline to produce synthetic dataset for marine visual understanding, which could be uesful for downstream tasks in this domain.
2. The automatic pipeline save the huge human cost to capture the marine data.
3. The overall pipeline is simple and easy to follow.

**Weaknesses:**

It is promising to utilize latest Vision-Language Models and Generative Models to construct the marine dataset. However, the diversity and quality of the dataset are not well proved.
1. For diversity: This work utilzes marine conception list to construct the whole dataset, which is promising. But the whole pipeline highly rely on Text-to-Image Model. It is trained on daily datasets that could be long-tail and not include some marine classes. Have the author checked the generated data of each categories? Consider list some examples and performance on uncommon classes with generated data.
2. For quality: In Table 2 and 3, the largest model is ResNet-50, which does not match current vision development. And the performance on small models cannot well prove the effectiveness and quality of the generated data. Consider conduct experiments on larger models like ViT-L and above.
3. How about the overall cost to construct the whole dataset? Consider provide a list that details the computation, token for GPT, and human cost for the whole pipeline. It is essential to measure the efficiency of the proposed pipeline.

**Questions:**

My main concerns are the diversity and quality of the dataset. Please refer to the weakness section.

---

### Official Review · Reviewer_y9UM · 2024-11-02

**Soundness:** 2
**Presentation:** 3
**Contribution:** 1
**Rating:** 3
**Confidence:** 4

**Summary:**

This paper proposes leveraging pre-trained generative models to synthesize training data for classification, segmentation, and question answering tasks related to marine visual problems. The authors generated four million marine images and demonstrated that using synthetic images can enhance vision performance in these areas. To improve the quality of data generation, the authors introduced a preference-picking strategy.

**Strengths:**

* The methodology presented in the paper is straightforward and easy to follow.
* The authors provide a new dataset focused on marine imagery, which has the potential to be highly beneficial for the marine research community.

**Weaknesses:**

1. The reviewer’s major concern is that this paper makes very limited contributions when viewed as a work in representation learning / machine learning / computer vision. The effectiveness of using diffusion-generated synthetic datasets to enhance performance in vision tasks has already been well-explored in existing studies (e.g., [A, B, C, D] for classification, [E, F, G] for segmentation). Considering the audience of ICLR, the reviewer doubts that this work will generate significant interest or provide new insights into learning or vision problems, as it is confined to marine-specific applications. The generalizability of this training strategy has already been validated in existing studies that use synthetic images to improve performance. This work may be better suited for a conference or journal focused on marine research.
2. There should be a quantitative analysis of the impact of the preference-based image picking strategy. While it may be redundant to demonstrate that this strategy improves overall image quality, understanding the extent of this improvement can be helpful.
3. Given the descriptive prompts in the dataset, the reviewer wonders why the authors chose to formalize the question-answering task as a binary classification problem (though the reviewer noticed a previous work also formalize it as a binary classificaiton, but that was on ImageNet dataset, which does not have descriptive prompts). Also, the types of questions are described in vague terms, making it difficult for readers to understand twhat questions have being asked.


[A] Is synthetic data from generative models ready for image recognition?

[B] Leaving Reality to Imagination: Robust Classification via Generated Datasets

[C] Synthetic Data from Diffusion Models Improves ImageNet Classification

[D] Enhance Image Classification via Inter-Class Image Mixup with Diffusion Model

[E] Improving Semantic Segmentation Models through Synthetic Data Generation via Diffusion Models

[F] Diffusion-based Data Augmentation for Nuclei Image Segmentation

[G] SatSynth: Augmenting Image-Mask Pairs through Diffusion Models for Aerial Semantic Segmentation

**Questions:**

The main reason for my rating is that the paper makes limited contributions in the field of representation learning / machine learning / computer vision (weakness 1). The use of diffusion-generated synthetic datasets for improving vision task performance has already been well-established in prior research. Given the scope of ICLR's audience, this work may not generate significant interest or provide new insights. Also, weaknesses 2 and 3, concerning the lack of quantitative analysis of the preference-based image picking strategy and the vague presentation of the question-answering task, further prevent the reviewer to give an acceptance.

---

### Official Review · Reviewer_aMPA · 2024-11-04

**Soundness:** 2
**Presentation:** 3
**Contribution:** 2
**Rating:** 5
**Confidence:** 4

**Summary:**

The paper proposes a synthetic data generation pipeline specifically fine-tuned to specifically generate marine images. The pipeline consists of a series of pre-trained foundation models namely an LLM (ChatGPT) to synthesize captions, a VLM (BLIP2) to generate captions for real marine images, and a text to image generator (StableDiffusion) to output the final synthetic images. The Stable Diffusion model is further fine-tuned on real marine images to be tailored towards generating higher quality marine synthetic datasets for the task at hand. The results show that combining the generated synthetic images with real images can improve baselines on tasks such as few-shot classification, and segmentation on the Sea-animal marine dataset.

**Strengths:**

- Although the use of synthetic data augmentation as a means for improving visual understanding has been previously explored in the general image understanding domain (see the following section for a few citations), its application to the specific domain of marine image understanding was unexplored.
- The proposed pipeline is simple to construct and relies on existing pre-trained models with minor changes (small fine-tuning required).
- The proposed approach is easy to understand and the paper is easy to follow.

**Weaknesses:**

- Claims

To me, the claims in the paper are not well backed in the current version of the submission. To be more specific, the title and the paper claims to be a "Never Ending" learning approach. However, by design, the method falls short on this claim. One important aspect of this claim is to be able to generalize to new concepts that come to existence over time. However, the synthetic data generation pipeline is relying on pre-trained/fine-tuned models that are trained on a training dataset frozen in time and not automatically getting updated to include new concepts. I also could not find an experiment showing generalization to new concepts that were not captured at the time of pre-training / fine-tuning the synthetic data generation pipeline. On the contrary, the experiments show that the performance improvement saturates after a relatively small number of synthetically generated images added to the training.  Moreover, throughout the paper there are claims that the method is a pure synthetic approach (e.g. Table 1 in the paper). However, the synthetic generation pipeline itself is relying on giant pre-training/fine-tuning real datasets.

- Experimental Setup:

The main contribution and message of the paper is that the NeMal's synthetic marine images / captions can improve models' performance on downstream marine tasks. One of my main concerns that makes it challenging to evaluate this claim is the internet data leakage into the synthetic data generation pipeline and consequently into the reported experiments. The proposed synthetic generation pipeline relies on pre-trained models (some with undisclosed training sets) that are very likely trained on additional marine related real data and categories / tasks that are similar to the test set that other baselines are not trained on. A more controlled experimental setup is necessary to verify this main claim (as an example see "Preventing Leakage Of Internet Data" section in reference [2] below).

- Additional related works:

I suggest the authors add a separate subsection on the related methods that use synthesis image generation for improving visual understanding on general visual tasks and also clarify the technical contributions over the existing works. Here are a few additional relevant works that authors can consider including:

[1] Synthetic Data from Diffusion Models Improves ImageNet Classification, Azizi et. al.

[2] Effective Data Augmentation With Diffusion Models, Trabucco et. al.

[3] DiffuseMix: Label-Preserving Data Augmentation with Diffusion Models, Islam et. al.

[4] CtrlSynth: Controllable Image Text Synthesis for Data-Efficient Multimodal Learning, Cao et. al.

**Questions:**

The authors can find additional comments and questions below.
1) One of my primary concerns is the potential data leakage in the synthetic generation pipeline itself. Particularly making it hard to determine if the improvement is coming simply because of having access to additional real data or the generalization of the synthetic data generation. Did you take any measures to mitigate data leakage?

2) Re the never learning claim in the paper, the experiments suggest that the learning saturates and mostly stops after generating a couple of thousand images. Is there an experimental validation that the never-ending learning occurs in practice? How does the method generalize to new concepts that are discovered after the synthetic data pre-training/fine-tuning is over? (e.g. how the result changes if one removes images/captions of a specific category that exists in the test set from the pre-training set of all the modules in the NeMal synthetic data generation pipeline (LLM, VLM, SD)  as well as its fine-tuning stage)

---

### Meta-Review · Area_Chair_MzCn · 2024-12-19

**Metareview:**

This paper proposes an approach to synthesizing marine images for marine visual understanding. All reviewers give negative scores with concerns on limited contribution, flawed claims and experiments, and invalidated quality and diversity of the dataset. The authors did not provide the rebuttal. So the area chair would recommend rejecting this paper and suggest the authors to improve the paper and presentation for future submissions.

**Additional Comments On Reviewer Discussion:**

No rebuttal provided by authors, so there's no further discussion.

---

### Decision · Program_Chairs · 2025-01-22

Reject